# Two Birds with One Stone: Neural Tangent Kernel for Efficient and Robust Gradual Domain Adaptation

## Abstract

Gradual Domain Adaptation (GDA) bridges large distribution shifts through intermediate domains, yet faces challenges in computational overhead and error accumulation. In view of these problems, we propose GradNTK, a novel framework to employ the Neural Tangent Kernel (NTK) as one stone to "hit" two birds of the efficiency and robust issues in GDA. On one hand, by exploiting the short-time dynamics of wide neural networks, GradNTK instantiates an NTK-induced Maximum Mean Discrepancy (MMD) as a differentiable discrepancy metric that enforces smooth transitions between adjacent domains while maintaining near-linear computational cost. On the other hand, the same NTK dynamics construct a sample reweighting function to weight source/target samples by their shift sensitivity, enabling curriculum-guided gradual adaptation while avoiding error accumulation. Experiments on Portraits, Rotated MNIST and CIFAR-100-C demonstrate superior performance (e.g., 95.1% on Rotated MNIST, 99.5% on Color-Shift MNIST), while reducing training time by 1.8× compared to prior GDA methods.

## 1 Introduction

Traditional machine learning theory typically assumes training and testing data share independent and identical distributions, a condition rarely met in real-world applications (Farahani et al., 2021) where the training and testing domain data usually have distribution shifts. Unsupervised Domain Adaptation (UDA) is proposed to reduce the distribution shifts by transferring knowledge from the labeled source domain to the unlabeled target domain (Farahani et al., 2021;

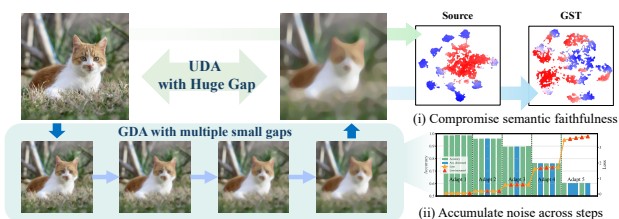

Figure 1: UDA directly aligns the two domains even the shifts are substantially large, while GDA decomposes the shifts into smaller steps for smoother adaptation. Right: (i) GDA (GST (Kumar et al., 2020) as a representative) causes class entanglement; (ii) In GDA, performance decays and loss grows when moving to the next domain—and even later epochs within a domain can harm performance.

Ben-David et al., 2006; Pan & Yang, 2009; Hoffman et al., 2018). However, UDA methods usually align the two domains directly(Pan & Yang, 2009), which induces negative transfer or even model collapse if the domain shifts are substantially large(Kang et al., 2019; Tang & Jia, 2020; Yang et al., 2020; Pan & Yang, 2009), as illustrated in Fig. 1.

To address these limitations, Gradual Domain Adaptation (GDA) has emerged as a more robust paradigm (Kumar et al., 2020). Instead of bridging source and target domains in a single step, GDA introduces a sequence of intermediate domains, allowing the model to adapt progressively through smoother transitions, as illustrated in fig. 1. Despite its effectiveness, existing GDA methods still face the efficiency and robustness challenges: (1) non-smooth inter-domain transitions under high computational costs (Saito et al., 2018; Pan & Yang, 2009), which is caused by inefficient alignment mechanisms; and (2) error accumulation (Lee et al., 2013; Zou et al., 2019; Arazo et al., 2020; Wang et al., 2020; Cicek & Soatto, 2019), where noise and misclassification of outliers propagate across domains, degrading the generalization of the model.

In this paper, we propose GradNTK, a novel GDA framework that leverages the Neural Tangent Kernel (NTK) to simultaneously tackle both issues in a unified and efficient framework. Drawing inspiration from recent advances in kernel methods and deep learning theory (Jacot et al., 2018), GradNTK employs an NTK-induced Maximum Mean Discrepancy (MMD) as a differentiable discrepancy metric (Cheng & Xie, 2021). This allows the model to enforce smoother transitions between adjacent domains while operating at a near-linear computational cost, which addresses the first challenge of efficiency and smoothness.

Moreover, we further exploit the short-time dynamics of wide neural networks under the NTK theory to derive an NTK-reweighting function, which weights source and target samples based on the shift sensitivity. This shifts-aware reweighting mechanism effectively reduces the influence of highly uncertain or noisy samples, thereby mitigating the second challenge of error propagation and accumulation throughout the adaptation process.

Overall, by integrating NTK-based distribution matching with curriculum-guided sample reweighting, GradNTK achieves highly efficient and robust training for GDA, hitting the two birds (efficiency and robustness issues) with one stone (NTK theory). The contributions are summarized as follows:

- We propose GradNTK, which optimizes a differentiable NTK-induced MMD metric to enforce smooth, near-linear transitions across intermediate domains.
- We exploit the short-time dynamics of a wide network to derive a differentiable NTK-reweighting function that produces sample weights for both source and target data, reducing error accumulation and enhancing adaptation stability.
- Extensive experiments on benchmarks including Rotated MNIST, Color-Shift MNIST, Portraits, and CIFAR-10-C/100-C demonstrate that GradNTK achieves superior performance while significantly reducing training time compared to existing methods.

## 2    RELATED WORK

**Gradual Domain Adaptation.**    Gradual domain adaptation (GDA) addresses large distribution shifts by leveraging intermediate domains between source and target (Kumar et al., 2020). Methods are typically categorized as interpolation-based, which construct virtual domains via linear or hybrid interpolation to smooth transitions (Wang et al., 2022; He et al., 2023; Sagawa & Hino, 2025), or self-training variants that incorporate adversarial regularization or sample selection (Zhang et al., 2021b; Shi & Liu, 2024). Foundational work established the first non-vacuous generalization bound and highlighted the importance of regularization (Kumar et al., 2020), followed by approaches for domain discovery (Chen & Chao, 2021), optimal sequencing (Wang et al., 2022), and virtual interpolation when intermediates are absent (Abnar et al., 2021).

**Neural Tangent Kernel.**    The double-descent phenomenon, initially suggested by classical bias–variance analysis (Geman et al., 1992; Belkin et al., 2019), has been empirically confirmed across vision and language tasks (Nakkiran et al., 2021; Zhang et al., 2021a). The neural tangent kernel (NTK) framework showed that gradient descent on infinitely wide networks is equivalent to kernel regression, yielding convergence and generalization guarantees (Jacot et al., 2018; Du et al., 2018; Arora et al., 2019b). Extensions of this perspective include distributional measures in RKHS (Cheng & Xie, 2021), multi-task transfer (Heiss et al., 2021), and domain adaptation via NTK-informed cross-domain regression (Wu et al., 2022).

## 3    METHOD

In this section, we derive the NTK-based formulas used in our method and analysis, and detail the GradNTK implementation.

### 3.1    NOTATION AND DEFINITIONS

**Domain and feature space.**    Let $\mathbb{X} \subseteq \mathbb{R}^{h \times w \times c}$ denote the space of images. We use a representation map $r(\cdot; \psi) : \mathbb{X} \to \mathbb{Z} \subseteq \mathbb{R}^d$ and a classifier $k(\cdot; \varphi) : \mathbb{Z} \to \mathbb{Y}$ with label space $\mathbb{Y} = \{1, 2, \ldots, C\}$.

Table 1: Frequently used symbols

| Symbol | Meaning |
|--------|---------|
| $\mathcal{D}_i$ | Distribution of domain $i$ ($i=0{:}n$) |
| $\mathcal{Z}_i$ | Feature distribution induced by $\mathcal{D}_i$ |
| $r(\cdot; \psi)$ | Feature extractor, $\mathbb{X} \to \mathbb{Z}$ |
| $k(\cdot; \varphi)$ | Label head (classifier), $\mathbb{Z} \to \mathbb{Y}$. |
| $f(\cdot; \theta)$ | Witness network; $g = f_T - f_0$. |
| $K, \Theta$ | Generic kernel; NTK of $f$ |

**Gradually shifting domains.** We consider $n+1$ related domains indexed by $0, \ldots, n$: domain 0 is the source, domain $n$ the target, and $1, \ldots, n-1$ form a smooth transition. Each domain $i$ has distribution $\mathcal{D}_i \in \mathcal{P}(\mathbb{X} \times \mathbb{Y})$ and induced feature distribution $\mathcal{Z}_i = (r_\psi \circ \pi_{\mathbb{X}})_{\#} \mathcal{D}_i \in \mathcal{P}(\mathbb{Z})$.

**Generic kernel and NTK.** We consider a wide witness network $f(\cdot; \theta)$ and write its centered output as $g(x) = f_T(x) - f_0(x)$. Let $K : \mathbb{Z} \times \mathbb{Z} \to \mathbb{R}$ be a positive semidefinite kernel with RKHS $\mathcal{H}_K$ and feature map $\phi_K$ such that $K(z, z') = \langle \phi_K(z), \phi_K(z') \rangle_{\mathcal{H}_K}$ (Gretton et al., 2012; Smola et al., 2007; Sriperumbudur et al., 2010b). For $f$, the Neural Tangent Kernel (NTK) at initialization $\theta_0$ is $\Theta(x, x') := \nabla_\theta f_{\theta_0}(x)^\top \nabla_\theta f_{\theta_0}(x')$.

### 3.2 NTK-Linearized Witness Network

We begin by recapping kernel MMD and its witness view, then linearize a parametric network in the infinite-width NTK regime to obtain a short-time NTK-induced alignment loss.

**Kernel MMD.** Let $K : \mathbb{Z} \times \mathbb{Z} \to \mathbb{R}$ be a positive semidefinite kernel with reproducing kernel Hilbert space (RKHS) $\mathcal{H}_K$ and feature map $\phi_K$. For distributions $\mathcal{X}$ and $\mathcal{Y}$ on $\mathbb{Z}$, their kernel mean embeddings are $\mu_{\mathcal{X}} := \mathbb{E}_{x \sim \mathcal{X}}[\phi_K(x)]$ and $\mu_{\mathcal{Y}} := \mathbb{E}_{y \sim \mathcal{Y}}[\phi_K(y)]$. The (population) squared kernel MMD (Gretton et al., 2012) is the squared RKHS distance between these embeddings:

$$\mathrm{MMD}_K^2(\mathcal{X}, \mathcal{Y}) := \left\| \mu_{\mathcal{X}} - \mu_{\mathcal{Y}} \right\|_{\mathcal{H}_K}^2. \tag{1}$$

Given finite samples $X = \{x_i\}_{i=1}^{|X|}$ and $Y = \{y_j\}_{j=1}^{|Y|}$ drawn from $\mathcal{X}$ and $\mathcal{Y}$, the unbiased empirical estimator of $\mathrm{MMD}_K^2$ is

$$\widehat{\mathrm{MMD}}_K^2(X, Y) = \frac{1}{|X|^2} \sum_{i=1}^{|X|} \sum_{i'=1}^{|X|} K(x_i, x_{i'}) - \frac{2}{|X||Y|} \sum_{i=1}^{|X|} \sum_{j=1}^{|Y|} K(x_i, y_j) + \frac{1}{|Y|^2} \sum_{j=1}^{|Y|} \sum_{j'=1}^{|Y|} K(y_j, y_{j'}), \tag{2}$$

which requires $O\big((|X| + |Y|)^2\big)$ kernel evaluations and becomes expensive on large datasets.

**Witness Function Representation.** The MMD between $\mathcal{X}$ and $\mathcal{Y}$ can be expressed as the integral probability metric (IPM) over the unit ball of $\mathcal{H}_K$ (Sriperumbudur et al., 2010a; Müller, 1997):

$$\mathrm{MMD}_K(\mathcal{X}, \mathcal{Y}) = \sup_{\substack{f \in \mathcal{H}_K \\ \|f\|_{\mathcal{H}_K} \leq 1}} \Big( \mathbb{E}_{x \sim \mathcal{X}}[f(x)] - \mathbb{E}_{y \sim \mathcal{Y}}[f(y)] \Big). \tag{3}$$

The discrepancy function is expressed as

$$\Delta_{\mathcal{X}}^{\mathcal{Y}}(f) := \mathbb{E}_{x \sim \mathcal{X}}[f(x)] - \mathbb{E}_{y \sim \mathcal{Y}}[f(y)]. \tag{4}$$

Then eq. (3) can be written as

$$\mathrm{MMD}_K(\mathcal{X}, \mathcal{Y}) = \sup_{\|f\|_{\mathcal{H}_K} \leq 1} \Delta_{\mathcal{X}}^{\mathcal{Y}}(f). \tag{5}$$

The maximizer (when $\mathcal{X} \neq \mathcal{Y}$) is proportional to the difference of the kernel mean embeddings, known as the *witness function* (Gretton et al., 2012; Muandet et al., 2017):

$$f^\star(\cdot) \propto \mu_{\mathcal{X}}(\cdot) - \mu_{\mathcal{Y}}(\cdot) = \int_{\mathbb{Z}} K(\cdot, x) \, d(\mathcal{X} - \mathcal{Y})(x), \tag{6}$$

where $\mu_{\mathcal{X}} := \int K(\cdot, x) \, d\mathcal{X}(x)$ and $\mu_{\mathcal{Y}} := \int K(\cdot, x) \, d\mathcal{Y}(x)$. Normalizing to unit RKHS norm gives (Gretton et al., 2012; Muandet et al., 2017)

$$f^\star = \frac{\mu_{\mathcal{X}} - \mu_{\mathcal{Y}}}{\|\mu_{\mathcal{X}} - \mu_{\mathcal{Y}}\|_{\mathcal{H}_K}}. \tag{7}$$

Substituting (7) into (4) yields $\Delta_{\mathcal{X}}^{\mathcal{Y}}(f^\star) = \mathrm{MMD}_K(\mathcal{X}, \mathcal{Y})$.

### 3.3 NTK Witness Functions Are MMD Estimators

Without the explicit computations in $\mathcal{H}_K$, we approximate the witness with a parametric family $\{f(\cdot;\theta) : \mathbb{Z} \to \mathbb{R}, \ \theta \in \Theta\}$. The *empirical discrepancy* induced by $f(\cdot;\theta)$ is (Gretton et al., 2012)

$$\widehat{\Delta}(f_\theta) := \int_{\mathbb{Z}} f(x;\theta)\, d(\hat{p} - \hat{q})(x) = \frac{1}{|X|} \sum_{i=1}^{|X|} f(x_i;\theta) - \frac{1}{|Y|} \sum_{j=1}^{|Y|} f(y_j;\theta). \tag{8}$$

Given that practical learning is implemented as loss minimization, we take the negative of (8):

$$\hat{L}(\theta) := -\widehat{\Delta}(f_\theta) = -\frac{1}{|X|} \sum_{i=1}^{|X|} f(x_i;\theta) + \frac{1}{|Y|} \sum_{j=1}^{|Y|} f(y_j;\theta). \tag{9}$$

Minimizing $\hat{L}(\theta)$ is therefore equivalent to *maximizing* the empirical discrepancy between the network outputs on $X$ and $Y$, yielding a data-driven approximation to the MMD witness function shaped by the inductive bias (and any regularization) of the chosen function class (Sriperumbudur et al., 2010a).

In the NTK (lazy-training) regime, training a sufficiently wide network on the signed witness loss in eq. (9) implicitly performs a kernel two–sample test. We only state the resulting properties here; detailed derivations are given in App. A.

**Lemma 1** (NTK Linearised Dynamics). *Let $f(x;\theta)$ be a width-$m$ network with NTK $\Theta_m(x,x')$. Under lazy training (width $m \to \infty$), gradient flow on any scalar loss $\mathcal{L}(\theta)$ yields, for all training times $t$ (Jacot et al., 2018; Lee et al., 2019; Arora et al., 2019b),*

$$\dot{f}_t(x) = -\sum_{i=1}^{n_{\mathrm{tr}}} \frac{\partial \ell}{\partial f}\big(f_t(x_i), y_i\big)\, \Theta(x, x_i), \tag{10}$$

*where $\ell$ is the per-example loss, $\Theta = \lim_{m\to\infty} \Theta_m$ is the limiting NTK, and $n_{\mathrm{tr}}$ denotes the number of training samples.*

**Lemma 2** (Kernel Expansion of the Contrastive Witness). *Choose signed targets $s_i = \frac{1}{|X|}\mathbf{1}_{\{x_i \in X\}} - \frac{1}{|Y|}\mathbf{1}_{\{x_i \in Y\}}$ and minimise $\hat{L}(\theta) = -\sum_i s_i f(x_i;\theta)$. Gradient flow produces (Arora et al., 2019a; Allen-Zhu et al., 2019), for any finite time $T$,*

$$g(x) := f_T(x) - f_0(x) = \sum_{i=1}^{n_{\mathrm{tr}}} \alpha_i(T)\, \Theta(x, x_i), \qquad \alpha_i(T) = c_T\, s_i, \tag{11}$$

*i.e. a signed kernel expansion over the empirical samples. Where $c_T$ is a scalar depending on training time summarizing accumulated step sizes.*

**Lemma 3** (Equivalence to Empirical MMD). *Let $\hat{p}, \hat{q}$ be the empirical measures on $X$ and $Y$. For the function $g$ in Lemma 2,*

$$\frac{1}{|X|}\sum_{x \in X} g(x) - \frac{1}{|Y|}\sum_{y \in Y} g(y) = c_T\, \widehat{\mathrm{MMD}}^2_\Theta(\hat{p}, \hat{q}), \tag{12}$$

*hence, $g$ is (up to the scalar $c_T$) the RKHS witness function for the empirical NTK-MMD (Cheng & Xie, 2021; Muandet et al., 2017).*

We validate the NTK-MMD in App. C, and formally state in App. B both its finite-width deviation bound (with an $O(m^{-1/2})$ convergence rate under standard wide-network assumptions) and its metric properties: $\mathrm{MMD}_\Theta(P,Q)$ is always nonnegative and becomes a proper metric (i.e., $\mathrm{MMD}_\Theta(P,Q) = 0$ iff $P = Q$) when the NTK kernel is characteristic, along with a discussion of these properties.

### 3.4 NTK-Reweighting Function

The previous lemmas show that the trained witness $g$ coincides with the empirical NTK-MMD witness (up to $c_T$), that is $g(z) \approx c_T\langle\phi_\Theta(z), \mu_{\hat{p}} - \mu_{\hat{q}}\rangle_{\mathcal{H}_\Theta}$. Here, $\mathcal{H}_\Theta$ denotes the RKHS induced

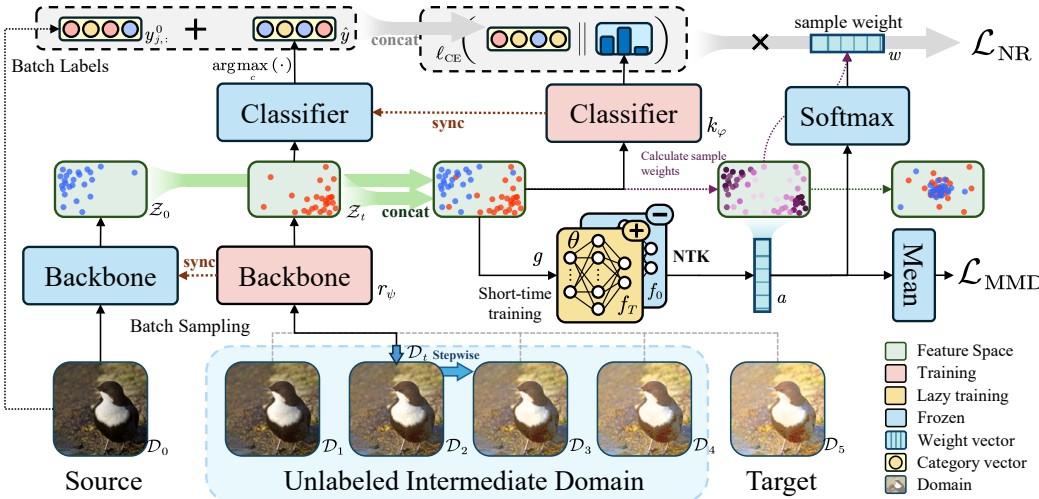

Figure 2: **Method overview of GradNTK.** A shared backbone $r_\psi$ encodes the source and current unlabeled batches into feature clouds $\mathcal{Z}_0$ and $\mathcal{Z}_t$. The yellow NTK block implements the NTK witness $g = f_T - f_0$ via short-time (lazy-training) evolution around $f_0$, mapping features to a scalar reweighting score $a(z)$; a temperature-scaled softmax produces per-sample weights $w(z)$ that gate the NTK-reweighting loss $\mathcal{L}_{\text{NR}}$ on the classifier. In parallel, the NTK-MMD loss $\mathcal{L}_{\text{MMD}}$ aligns source and unlabeled features, and lightweight source replay preserves class semantics. Because $g$ lies in the RKHS of the NTK $\Theta$, $a(z)$ is smooth in feature space and the pipeline remains end-to-end differentiable.

by the NTK $\Theta$, $\phi_\Theta$ is its implicit feature map. We now use this function to select informative samples from an unlabeled pool $U$ to reduce domain discrepancy (Cheng & Xie, 2021). The signed projection distance is rewritten as

$$d(z) = \frac{g(z)}{c_T \|\Delta\mu\|_{\mathcal{H}_\Theta}}, \tag{13}$$

where $|d(z)|$ measures discrimination of $z$ between domains. The reweighting score balances this with the gradient magnitude:

$$a(z) = \frac{|g(z)|^\alpha}{\sqrt{\Theta(z,z) + \epsilon}}, \tag{14}$$

where $\epsilon > 0$ stabilizes the denominator, with $\alpha = 1$ in our default experiments. Intuitively, samples that reside in regions of large source–target discrepancy should contribute more strongly to adaptation (Sugiyama et al., 2007; Sener & Savarese, 2017; Kirsch et al., 2019). These scores are normalized via a temperature-controlled softmax (Hinton et al., 2015; Guo et al., 2017) as follows:

$$w(z) = \frac{\exp(a(z)/\tau)}{\sum_{z' \in \mathcal{B}} \exp(a(z')/\tau)}, \tag{15}$$

where $\tau > 0$ controls the sharpness of the distribution. A first-order perturbation analysis of the batch MMD objective (see Sec. 3.3) shows that adding a sample $z$ with large $|g(z)|$ is predicted to yield a larger decrease in MMD, justifying Lemma 3.

A formal proposition establishing the target-risk bound and the first-order optimality interpretation of the NTK-based weights is provided in App. B, and we further verified the spatial significance of the NTK-reweighting function in Apps. C and F.

### 3.5 GRADNTK FRAMEWORK

As illustrated in fig. 2, GradNTK operates in an online gradual domain adaptation setting (Bobu et al., 2018; Wang et al., 2019). At adaptation step $t$, we have access to a labeled *source* dataset $D_0$

sampled i.i.d. from a (fixed) source distribution $\mathcal{D}_0$ and an *unlabeled* mini-batch stream from the *current* target domain with underlying distribution $\mathcal{D}_t$. We form mini-batches

$$D_0 = \bigsqcup_{j=1}^{n} B_j^0, \qquad D_t = \bigsqcup_{i=1}^{m} B_i^t. \tag{16}$$

where $\bigsqcup$ denotes disjoint union, $B_j^0 = \{(x_{jk}^0, y_{jk}^0)\}_{k=1}^{b_0}$ contains $b_0$ i.i.d. labeled samples from $\mathcal{D}_0$, and $B_i^t = \{x_{ik}^t\}_{k=1}^{b_t}$ contains $b_t$ i.i.d. unlabeled samples from $\mathcal{D}_t$. We write $b_0 = |B_j^0|$ and $b_t = |B_i^t|$ for the batch sizes.

For the convenience of description, we decompose the model into a feature extractor $r(\cdot; \psi)$ and a *classifier head* $k(\cdot; \varphi)$ producing logits in $\mathbb{R}^C$ for $C$ classes:

$$h(x; \psi, \varphi) := k(r(x; \psi); \varphi), \quad p(y = c \mid x; \psi, \varphi) = \text{softmax}_c(h(x; \psi, \varphi)). \tag{17}$$

We abbreviate $r(x) \equiv r(x; \psi)$ and $h(x) \equiv h(x; \psi, \varphi)$.

**Empirical linearized MMD optimization.** Given a paired mini-batch $(B_j^0, B_i^t)$, we set $K = \Theta$ and form the empirical witness:

$$\hat{g}_{j,i}(z; \psi) := \frac{1}{b_0} \sum_{(x,y) \in B_j^0} K(r(x; \psi), z) - \frac{1}{b_t} \sum_{x \in B_i^t} K(r(x; \psi), z). \tag{18}$$

To optimize the linearized MMD objective in eq. (12), we minimize the mean witness value within the paired batches:

$$\mathcal{L}_{\text{MMD}}(\psi; B_j^0, B_i^t) = \frac{1}{b_0} \sum_{(x,y) \in B_j^0} \hat{g}_{j,i}(r(x; \psi)) - \frac{1}{b_t} \sum_{x \in B_i^t} \hat{g}_{j,i}(r(x; \psi)). \tag{19}$$

Minimizing eq. (19) enforces representation alignment between source and the intermediate domain samples (Gretton et al., 2012; Long et al., 2015). (See eq. (12) and derivation in Sec. 3.3 for the equivalence under the NTK linearization.)

**Dynamic pseudo-labels.** We require labels to couple the classification objective with domain alignment. For source samples $(x, y) \in B_j^0$ we use their ground-truth labels $y$. For target samples $x \in B_i^t$ we apply *online pseudo-labeling* with confidence filtering. We define

$$\tilde{p}(x) = \text{softmax}(h(x; \psi, \varphi)), \quad \hat{y}(x) = \arg\max_c \tilde{p}_c(x), \quad \text{conf}(x) := \tilde{p}_{(1)}(x) - \tilde{p}_{(2)}(x), \tag{20}$$

where $(\psi, \varphi)$ are the parameters before updating on the current target batch, $\tilde{p}^{(1)}(x)$ and $\tilde{p}^{(2)}(x)$ denote the largest and second-largest entries of $\tilde{p}(x)$, and $\text{conf}(\cdot)$ is confidence. We exclude the bottom $q$-quantile of target samples by confidence.

We concatenate the labeled source batch and the retained, pseudo-labeled target batch $\widetilde{B}_i^t \subseteq B_i^t$:

$$B_c := B_j^0 \sqcup \{(x_{ik}^t, \hat{y}(x_{ik}^t))\}_{k=1}^{|\widetilde{B}_i^t|}. \tag{21}$$

Unlike generalized one-shot domain adaptation schemes that generate a fixed set of pseudo-labels which may entrench early misclassifications, we update pseudo-labels at each mini-batch using the most recent model parameters (Lee et al., 2013; Zou et al., 2018; Sohn et al., 2020). Thus, mislabeled target samples can be corrected as the model improves, limiting error accumulation and label drift across training iterations.

**NTK-reweighting supervised objective.** Given $(B_c, \hat{y}_c)$ from (21), we perform a NTK-reweighting update by minimizing the weighted cross-entropy

$$\mathcal{L}_{\text{NR}}(\psi, \varphi; B_c) = \sum_{(x,y) \in B_c} w(x) \, \ell_{\text{CE}}(h(x; \psi, \varphi), y), \tag{22}$$

where $\ell_{\text{CE}}$ is the standard (multi-class) cross-entropy and $w(x)$ is given by (15). Since each update mixes source data, the classifier is regularly re-anchored to the source label semantics. This continual *source rehearsal* mitigates catastrophic forgetting and reduces the risk of representation drift or class collapse while still steering $r(\cdot; \psi)$ toward a representation that narrows the source–target discrepancy (French, 1999; Li & Hoiem, 2017; Chaudhry et al., 2019).

**Overall update for step $t$.** For each incoming target batch $B_i^t$ we: (1) sample a contemporaneous source batch $B_j^0$; (2) compute the mini-batch witness $\hat{g}_{j,i}$; (3) pseudo-label and confidence-filter $B_i^t$; (4) build the combined labeled batch $B_c$; and (5) take gradient steps on the composite objective

$$\mathcal{L}_{\text{step-}t} = \lambda_{\text{MMD}} \, \mathcal{L}_{\text{MMD}}(\psi; B_j^0, B_i^t) + \mathcal{L}_{\text{NR}}(\psi, \varphi; B_c), \tag{23}$$

with trade-off weights $\lambda_{\text{MMD}} \geq 0$. The pseudocode for the framework can be found in app. H.

## 4 EXPERIMENTS

In this section, we present quantitative and qualitative comparisons demonstrating the efficacy of our approach, followed by ablation studies that isolate the contribution of each component.

### 4.1 SETUP AND DATASETS

We follow the classic GDA protocol with progressive self-training over an ordered sequence of intermediate domains. All methods are initialized from the same classifier pre-trained on the source domain for a fair comparison. All results report the mean over five independent runs. Following importance-weighted cross-validation (IWCV) (Sugiyama et al., 2007), we set the witness-network width to 512, the alignment weight to $\lambda_{\text{MMD}} = 1$, and the pseudo-label filtering quantile to $q = 0.10$; unless stated otherwise, these choices are fixed across datasets. Sensitivity to these choices is studied in the ablation analysis (see table 7).

**Standard benchmark.** Rotated MNIST is built from MNIST (Deng, 2012) with 50k source images at $0°$ and 50k target images at $45°$. Intermediate domains linearly interpolate the rotation angles. Color-Shift MNIST follows He et al. (2023): source images are normalized to $[0, 1]$, targets shifted to $[1, 2]$, and intermediates are linear color-intensity interpolations. Portraits (Ginosar et al., 2015) are chronologically divided into 9 temporal domains (1905–2013), each with 2,000 images (Kumar et al., 2020). The first and last act as source and target; images are resized to $32 \times 32$ pixels.

**Corruption benchmarks.** To test under realistic high-severity shifts, we evaluate on CIFAR-10-C, CIFAR-100-C (Hendrycks & Dietterich, 2019), which apply 15 corruption types at 5 severities to CIFAR (Krizhevsky et al., 2009). We treat clean training data as the source, severities 1–4 as intermediate domains, and severity 5 as the target; we verify progressiveness in App. D.

**Baselines & backbones.** We compare GradNTK against GST (Kumar et al., 2020) and GOAT (He et al., 2023) across three task families as follows: (i) Rotated MNIST, Color-Shift MNIST, and Portraits with three backbones—a small CNN (three 32-channel convolutional layers), ResNet-18 (He et al., 2016), and VGG11 (Simonyan & Zisserman, 2014); (ii) CIFAR-10-C with WideResNet-28 (Zagoruyko & Komodakis, 2016); and (iii) CIFAR-100-C with ResNeXt-29 (Xie et al., 2017).

### 4.2 RESULTS

We present our experimental results across synthetic, real, and corruption-robustness tasks.

Table 2: Comparison of domain adaptation methods on 3 GDA datasets.

| Methods | Gradual | Rotated MNIST | Color-Shift MNIST | Portraits |
|---|---|---|---|---|
| DANN (Ganin et al., 2016) | ✗ | 44.2 | 56.5 | 73.8 |
| DeepCoral (Sun & Saenko, 2016) | ✗ | 49.6 | 63.5 | 71.9 |
| DeepJDOT (Damodaran et al., 2018) | ✗ | 51.6 | 65.8 | 72.5 |
| GST (Kumar et al., 2020) (ICML'20) | ✓ | 83.8 | 74.0 | 82.6 |
| IDOL (Chen & Chao, 2021) (NeurIPS'21) | ✓ | 87.5 | - | 85.5 |
| AGST (Zhou et al., 2022) (IEEE'22) | ✓ | 76.2 | - | 77.6 |
| GGF (Zhuang et al., 2024) (ICLR'24) | ✓ | 67.7 | - | **86.2** |
| GOAT (He et al., 2023) (JMLR'24) | ✓ | 86.4 | 91.8 | 83.6 |
| DRO (Najafi et al., 2024) (NeurIPS'24) | ✓ | 53.2 | - | - |
| AST (Shi & Liu, 2024) (NeurIPS'24) | ✓ | 90.6 | - | 84.8 |
| CNF (Sagawa & Hino, 2025) (Neural Computation'25) | ✓ | 62.6 | - | 84.6 |
| GradNTK (Ours) | ✓ | **95.1** | **99.5** | 84.3 |

Table 3: Accuracy comparison on base three vision datasets using a shared source-domain pretrained classifier, evaluated with three backbones and 2–6 given domains.

| Dataset | Methods | CNN | | | | | ResNet | | | | | VGG | | | | | Mean |
|---|---|---|---|---|---|---|---|---|---|---|---|---|---|---|---|---|---|
| | | 2 | 3 | 4 | 5 | 6 | 2 | 3 | 4 | 5 | 6 | 2 | 3 | 4 | 5 | 6 | |
| Rotated MNIST | GST Kumar et al. (2020) | 54.7 | 57.1 | 58.6 | 60.1 | 60.3 | 71.0 | 70.8 | 70.9 | 71.3 | 71.4 | 67.6 | 69.7 | **71.3** | **72.7** | **73.7** | 66.7 |
| | GOAT He et al. (2023) | 54.2 | 58.6 | 61.1 | 64.9 | 67.9 | 71.6 | 71.8 | 72.0 | 72.0 | 72.2 | 67.6 | 68.4 | 69.1 | 69.5 | 70.2 | 67.4 |
| | GradNTK (ours) | **56.2** | **67.0** | **70.0** | **76.4** | **80.7** | **90.5** | **94.0** | **94.7** | **94.9** | **95.1** | **70.7** | **70.0** | 71.0 | 69.9 | 68.9 | **77.9** |
| Color-Shift MNIST | GST Kumar et al. (2020) | 71.5 | 66.6 | 70.0 | 72.6 | 77.9 | 95.9 | 97.0 | 99.0 | **99.7** | **99.7** | 89.6 | 94.1 | 97.1 | 98.2 | 99.0 | 88.5 |
| | GOAT He et al. (2023) | 84.0 | **94.0** | **96.9** | **97.7** | **98.0** | 94.5 | 94.9 | 95.1 | 95.3 | 95.3 | 90.7 | 92.7 | 93.6 | 94.2 | 94.8 | 94.1 |
| | GradNTK (ours) | **92.3** | 92.5 | 92.1 | 91.0 | 92.1 | **99.2** | **99.2** | **99.2** | 99.2 | 99.1 | **99.3** | **99.3** | **99.4** | **99.4** | **99.5** | **96.9** |
| Portraits | GST Kumar et al. (2020) | 70.2 | **72.1** | 72.0 | 73.2 | 72.3 | 73.0 | 73.5 | 73.7 | 73.7 | 73.1 | 64.9 | 66.0 | **67.7** | **68.9** | **68.8** | 70.9 |
| | GOAT He et al. (2023) | 71.1 | 71.7 | 71.9 | 72.6 | **72.9** | 71.6 | 72.7 | 73.0 | 73.1 | 73.6 | 65.7 | 65.7 | 65.6 | 65.7 | 65.6 | 70.2 |
| | GradNTK (ours) | **71.5** | 72.0 | **74.0** | **74.9** | **77.4** | **80.2** | **81.3** | **82.9** | **83.6** | **84.3** | **70.9** | **67.0** | 62.5 | 62.9 | 67.4 | **73.5** |

Table 4: Classification **error rates** (%) at severity 5 on CIFAR-10-C and CIFAR-100-C. *Source* is no adapt model; BN-1 and TENT-co. are TTA methods; GST and GOAT are GDA baselines.

| | Method | Gradual | gaussian | shot | impulse | defocus | glass | motion | zoom | snow | frost | fog | brightness | contrast | elastic | pixelate | jpeg | Mean |
|---|---|---|---|---|---|---|---|---|---|---|---|---|---|---|---|---|---|---|
| CIFAR-10-C | Source | ✗ | 72.3 | 65.7 | 72.9 | 46.9 | 54.3 | 34.8 | 42.0 | 25.1 | 41.3 | 26.0 | 9.3 | 46.7 | 26.6 | 58.5 | 30.3 | 43.5 |
| | BN-1 (Li et al., 2016) | ✗ | 28.1 | 26.1 | 36.3 | 12.8 | 35.3 | 14.2 | 12.1 | 17.3 | 17.4 | 15.3 | 8.4 | 12.6 | 23.8 | 19.7 | 27.3 | 20.4 |
| | TENT-co. (Wang et al., 2020) | ✗ | **24.8** | **20.6** | 28.6 | **14.4** | 31.1 | 16.5 | 14.1 | 19.1 | 18.6 | 18.6 | 12.2 | 20.3 | 25.7 | 20.8 | 24.9 | 20.7 |
| | GST Kumar et al. (2020) | ✓ | 50.0 | 43.9 | 50.3 | 20.6 | 51.2 | 17.2 | 16.7 | 17.5 | 24.3 | 17.5 | 6.9 | 13.2 | 24.9 | 39.9 | 26.6 | 28.1 |
| | GOAT He et al. (2023) | ✓ | 72.7 | 65.7 | 73.0 | 46.7 | 54.5 | 34.3 | 41.5 | 24.9 | 41.0 | 26.0 | 9.3 | 46.6 | 26.4 | 58.1 | 30.2 | 43.4 |
| | GradNTK (ours) | ✓ | 29.3 | 26.4 | 33.0 | **14.4** | **28.4** | **11.8** | **11.1** | **11.1** | **13.0** | **10.6** | **2.6** | **10.7** | **19.7** | **18.6** | **20.1** | **17.4** |
| CIFAR-100-C | Source | ✗ | 73.0 | 68.0 | 39.4 | 29.3 | 54.1 | 30.8 | 28.8 | 39.5 | 45.8 | 50.3 | 29.5 | 55.1 | 37.2 | 74.7 | 41.2 | 46.4 |
| | BN-1 (Li et al., 2016) | ✗ | 42.1 | 40.7 | 42.7 | 27.6 | 41.9 | 29.7 | 27.9 | 34.9 | 35.0 | 41.5 | 26.5 | 30.3 | 35.7 | 32.9 | 41.2 | 35.4 |
| | TENT-co. (Wang et al., 2020) | ✗ | **37.2** | 35.8 | 41.7 | 37.9 | 51.2 | 48.3 | 48.5 | 58.4 | 63.7 | 71.1 | 70.4 | 82.3 | 88.0 | 88.5 | 90.4 | 60.9 |
| | GST Kumar et al. (2020) | ✓ | 49.8 | 56.7 | 32.3 | 22.5 | 41.6 | 25.0 | 23.3 | 30.3 | 32.2 | 38.1 | 22.1 | 27.0 | 33.1 | 40.8 | 35.8 | 33.3 |
| | GOAT He et al. (2023) | ✓ | 73.4 | 67.9 | 39.1 | 28.7 | 53.8 | 30.2 | 28.7 | 39.3 | 45.7 | 50.0 | 29.4 | 53.7 | 36.8 | 74.3 | 41.2 | 46.2 |
| | GradNTK (ours) | ✓ | 39.0 | **34.3** | **23.2** | **11.6** | **31.4** | **16.1** | **13.5** | **20.0** | **22.0** | **29.5** | **7.5** | **23.4** | **24.1** | **29.1** | **28.7** | **23.6** |

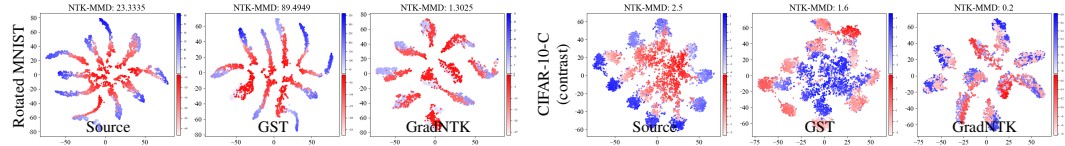

Figure 3: t-SNE visualizations of feature embeddings before (Source) and after adaptation with GST and the proposed GradNTK. Blue and red dots denote source- and target-domain samples, respectively; darker hues indicate a higher value of the NTK-reweighting function. GradNTK yields tighter class-wise structure, better source–target overlap, and lower NTK-MMD than GST, indicating more effective alignment.

**Main GDA comparison.** On the three canonical GDA datasets (Table 2), GradNTK attains state-of-the-art accuracy on *Rotated MNIST* (95.1%) and *Color-Shift MNIST* (99.5%). On *Portraits*, it is competitive; we trace the shortfall to the shift being concentrated in the final two temporal domains, where source–target consistency is weak, making our source-replay component bias the model toward the source and dampen target adaptation. We verified this diagnosis in App. C.

**Effect of intermediate domains and backbones.** Varying the number of given domains and backbone capacity (Table 3) shows consistent gains for GradNTK: on *Rotated MNIST* with a small CNN, accuracy rises from 56.2% (2 domains) to 80.7% (6 domains), and with ResNet-18 reaches 95.1% for 4–6 domains; on *Color-Shift MNIST* it remains ≥99.1% with ResNet/VGG; on *Portraits*, Grad-NTK yields the best mean across backbones.

**Corruption benchmarks.** At severity 5 on CIFAR-10-C/100-C (Table 4), GradNTK achieves the lowest mean error among the GDA baselines and performs favorably compared to standard streaming TTA baselines under their more restricted information setting. These gains support the design choice of coupling short-time NTK-MMD alignment with dynamic pseudo-label refinement and lightweight source replay. Sensitivity to the number of given domains is provided in App. E.

**Feature-space analysis.** t-SNE visualizations (fig. 3) indicate that GradNTK tightens class clusters and aligns source/target manifolds more effectively than GST; high NTK-reweighting values (darker) samples concentrate within class-consistent regions after adaptation, implying that the kernel-aware updates focus on the most uncertain, hardest-to-align regions. Additional plots are provided in App. F.

## 4.3 ABLATION STUDY

Our ablation analysis examines two factors: component-wise contributions and the influence of model capacity and hyperparameter choices.

Table 5: Efficiency analysis of GradNTK with a ResNet-18 backbone. We report target-domain accuracy (%) and peak GPU memory.

| Model/Experiment Setting | Rotated MNIST | | | | | Mean | Max Memory |
|---|---|---|---|---|---|---|---|
| | 2 | 3 | 4 | 5 | 6 | | |
| GradNTK | 90.5 | **94.0** | **94.7** | **94.9** | **95.1** | **93.8** | 1608MB |
| GradNTK (batch size 256) | **92.2** | 93.1 | 93.7 | 93.7 | 93.9 | 93.2 | **1020MB** |
| w/o NTK-reweighting | 90.1 | 93.6 | 94.2 | 94.5 | 94.7 | 93.4 | 1608MB |
| w/o NTK-MMD | 74.5 | 74.4 | 76.5 | 77.0 | 77.4 | 76.0 | 1598MB |
| w/o NTK-MMD w/ MMD | 82.9 | 87.0 | 87.7 | 88.5 | 88.8 | 87.0 | 42310MB |
| w/o NTK-MMD w/ MMD (batch size 256) | 82.9 | 86.2 | 87.0 | 88.0 | 88.4 | 86.5 | 3328MB |

Table 6: Robustness ablation on CIFAR-10-C. The table reports accuracy (%) for each corruption and the mean. Variants remove components as indicated.

| Model/Experiment Setting | gaussian | shot | impulse | defocus | glass | motion | zoom | snow | frost | fog | brightness | contrast | elastic | pixelate | jpeg | Mean |
|---|---|---|---|---|---|---|---|---|---|---|---|---|---|---|---|---|
| GradNTK | **70.7** | **73.6** | 67.0 | **85.6** | **71.6** | **88.2** | **88.9** | **88.9** | **87.0** | **89.4** | **97.4** | **89.3** | **80.3** | **81.4** | 79.9 | **82.6** |
| w/o NTK-reweighting | 69.5 | 72.5 | 65.1 | 85.0 | 70.2 | 87.4 | 88.4 | 87.8 | 85.9 | 88.8 | 97.2 | 88.9 | 80.1 | 78.4 | 78.9 | 81.6 |
| w/o Source Label | 70.0 | 70.8 | 67.7 | 85.5 | 71.2 | 87.2 | 88.9 | 86.2 | 85.7 | 87.4 | 92.9 | 86.2 | 79.9 | 81.3 | 80.5 | 81.4 |
| w/o NTK-MMD | 68.4 | 71.6 | 64.5 | 83.2 | 67.9 | 86.6 | 87.7 | 87.6 | 85.5 | 88.0 | 97.2 | 88.6 | 79.3 | 75.5 | 78.1 | 80.6 |

Table 7: Capacity and hyperparameter ablation on Rotated MNIST. Target-domain accuracy (%) as a function of NTK width (neurons) and alignment weight $\lambda_{\text{MMD}}$; all other settings fixed.

| Rotated MNIST | # Neurons | | | | | $\lambda_{\text{MMD}}$ | | | | |
|---|---|---|---|---|---|---|---|---|---|---|
| + ResNet | 64 | 128 | 256 | 512 | 1024 | 0.25 | 0.5 | 1 | 2 | 4 |
| GradNTK | 93.9 | 94.2 | 94.6 | **94.7** | 94.7 | 94.2 | 94.4 | **94.7** | 94.6 | 94.2 |

**Component contributions.** With a ResNet-18 backbone and 2–6 given domains, we compare the full model to controlled variants (Table 5): (i) *w/o NTK-MMD* drops the alignment term; (ii) *w/o NTK-MMD w/ MMD* replaces NTK-MMD by Gaussian-kernel MMD at the same weight; (iii) *w/o NTK-reweighting* disables NTK-reweighting (uniform sampling). We also report on a smaller batch size of 256 to investigate the accuracy. As summarized in Table 5, removing the alignment term markedly hurts performance, while substituting Gaussian MMD recovers only part of the gap and inflates memory to 42.3 GB ($\times 26$), highlighting the effectiveness and efficiency of the short-time NTK formulation. For details on the advantages regarding method runtime and memory usage, see App. G.

**Corruption-robustness ablation.** Using WideResNet-28 and 6 given domains, the model is progressively exposed to severities 1–4 across all 15 corruption types and evaluated at severity 5 (Table 6). We re-use the component variants above and report per-corruption and mean accuracies. As shown in Table 6, the full GradNTK attains the best mean accuracy (82.6%), outperforming *w/o utility weighting* (81.6, +1.0), *w/o NTK–MMD* (80.6, +2.0), and *w/o Source Label* (81.4, +1.2). GradNTK leads on 13/15 corruptions, with only marginal deficits on *impulse* and *jpeg*, indicating that short-time NTK alignment is the primary drivers of robustness.

**NTK Capacity & hyperparameters.** On Rotated MNIST with ResNet-18 and four given domains, we vary the NTK width and the alignment weight $\lambda_{\text{MMD}}$ while keeping all other settings

fixed (Table 7). Accuracy increases with width and saturates at $\approx 94.7\%$ once the width $\geq 256$, indicating diminishing returns beyond moderate capacity. The method is also insensitive to the alignment weight: $\lambda_{\mathrm{MMD}} \in [0.25, 2]$ yields 94.2–94.7% (peaking at $\lambda=1$) with only slight degradation at $\lambda=4$. Overall, GradNTK does not require fine-grained tuning of capacity or $\lambda_{\mathrm{MMD}}$.

## 5 CONCLUSION

We introduce the neural tangent kernel to gradual domain adaptation, using it both as a principled divergence metric for seamless source-to-target alignment and as the engine of an NTK-reweighting function that enables sample-efficient reweighting. This dual NTK role yields a memory-light, curriculum-guided procedure that consistently surpasses GDA baselines on the gradual benchmarks and performs favorably compared to standard streaming TTA baselines on CIFAR-10-C and CIFAR-100-C. These results establish a strong baseline for future work on more complex domain shifts.

### REPRODUCIBILITY STATEMENT

We took several steps to make our results easy to replicate. The method is fully specified in Sec. 3 with precise notation (table 1) and objectives, including the MMD witness formulation and its empirical expansion (eqs. (5) and (12)), alongside NTK-based lemmas whose proofs are deferred to App. A. The training pipeline is summarized in fig. 2 and alg. 1, and the step-wise update is given explicitly by eq. (23); these components together define the implementation contract for our algorithm. Experimental setup details are provided in Sec. 4.1, covering datasets and protocols for Rotated-MNIST, Color-Shift MNIST, Portraits, and CIFAR-10-C/100-C (where severities 1–4 serve as intermediates and severity 5 as target), the compared baselines/backbones, default hyperparameters (NTK width 512, $\lambda_{\mathrm{MMD}} = 1$, confidence-filtering quantile q = 0.10), and that all reported numbers are means over five independent runs. We assess robustness and component contributions via ablations (tables 5 and 6) and report capacity/hyperparameter sensitivity (table 7), which collectively indicate which ingredients and settings are necessary to match our results. Finally, we include diagnostic checks and additional analyses: validation of the NTK-MMD estimator and NTK-reweighting behavior (Apps. C and F), verification that the corruption path is progressive (App. D), and sensitivity to the number of given domains (App. E). Together, these referenced sections provide the information needed to reproduce training, evaluation, and analyses.

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

# A    DERIVATION DETAILS FOR SECTION 3.3

## A.1    PROOF OF LEMMA 1

We now connect the parametric witness loss in (9) to a kernel MMD in the *Neural Tangent Kernel* (NTK) regime. Consider a sufficiently wide neural network $f(x; \theta)$ initialized at parameters $\theta_0$. Following standard NTK analyses (Jacot et al., 2018), for small parameter displacements we linearize the network output:

$$f_t(x) \approx f_0(x) + \nabla_\theta f_0(x)^\top (\theta_t - \theta_0), \tag{24}$$

where $t$ indexes training steps and $f_0(\cdot) = f(\cdot; \theta_0)$.

Let $\mathcal{L}(\theta)$ be any scalar training objective (we will shortly specialize to the parametric witness loss). A single gradient descent step with learning rate $\eta > 0$ gives $\theta_{t+1} = \theta_t - \eta \nabla_\theta \mathcal{L}(\theta_t)$. Insert this into the linearization to obtain

$$\begin{aligned} f_{t+1}(x) &\approx f_t(x) + \nabla_\theta f_0(x)^\top (\theta_{t+1} - \theta_t) \\ &= f_t(x) - \eta \nabla_\theta f_0(x)^\top \nabla_\theta \mathcal{L}(\theta_t). \end{aligned} \tag{25}$$

Now suppose $\mathcal{L}$ decomposes over training examples $\{(x_i, y_i)\}_{i=1}^{n_{\mathrm{tr}}}$ as $\mathcal{L}(\theta) = \sum_{i=1}^{n_{\mathrm{tr}}} \ell(f(x_i; \theta), y_i)$, where $y_i$ are (possibly pseudo) targets. Then, by the chain rule, we have

$$\nabla_\theta \mathcal{L}(\theta_t) = \sum_{i=1}^{n_{\mathrm{tr}}} \frac{\partial \ell}{\partial f}(f_t(x_i), y_i) \, \nabla_\theta f_t(x_i). \tag{26}$$

Under the *lazy training* assumption (outputs remain close to initialization), replace $\nabla_\theta f_t(\cdot)$ by $\nabla_\theta f_0(\cdot)$ to obtain

$$f_{t+1}(x) \approx f_t(x) - \eta \sum_{i=1}^{n_{\mathrm{tr}}} \frac{\partial \ell}{\partial f}(f_t(x_i), y_i) \, \nabla_\theta f_0(x)^\top \nabla_\theta f_0(x_i). \tag{27}$$

Define the (time-0) Neural Tangent Kernel

$$\Theta(x, x') := \nabla_\theta f_0(x)^\top \nabla_\theta f_0(x'). \tag{28}$$

Then the update rule becomes

$$f_{t+1}(x) \approx f_t(x) - \eta \sum_{i=1}^{n_{\mathrm{tr}}} \frac{\partial \ell}{\partial f}(f_t(x_i), y_i) \, \Theta(x, x_i). \tag{29}$$

Iterating (29) (or passing to continuous-time gradient flow and solving the resulting linear ODE) yields that the network output at training time $T$ can be written as a *kernel expansion* over the training inputs:

$$f_T(x) \approx f_0(x) + \sum_{i=1}^{n_{\mathrm{tr}}} \alpha_i \, \Theta(x, x_i), \tag{30}$$

with coefficients $\alpha_i$ determined by the training signal $\frac{\partial \ell}{\partial f}(f_t(x_i), y_i)$ accumulated over time (and by any explicit regularization).

## A.2    PROOF OF LEMMA 2

To recover the parametric witness objective (9), we construct a *contrastive* training set by assigning *signed pseudo-labels* that encode membership in the two empirical samples $X$ and $Y$. Specifically, set

$$s_i = \begin{cases} +\frac{1}{|X|}, & \text{if } x_i \in X, \\ -\frac{1}{|Y|}, & \text{if } x_i \in Y, \end{cases}$$

and define the loss

$$\hat{L}(\theta) = -\sum_{i=1}^{n_{\mathrm{tr}}} s_i \, f(x_i; \theta) = -\frac{1}{|X|} \sum_{x_i \in X} f(x_i; \theta) + \frac{1}{|Y|} \sum_{y_j \in Y} f(y_j; \theta), \tag{31}$$

which is exactly (9).[1]

---

[1] One may view this as a linear loss (or the infinitesimal first-order part of a logistic / cross-entropy loss) whose gradient directions are aligned with the signed membership indicators $s_i$.

The gradient of $\hat{L}$ with respect to $\theta$ is

$$\nabla_\theta \hat{L}(\theta) = -\sum_{i=1}^{n_{\mathrm{tr}}} s_i \, \nabla_\theta f(x_i; \theta). \tag{32}$$

Applying the NTK linearization as above (and again freezing Jacobians at initialization) gives the functional update

$$f_{t+1}(x) \approx f_t(x) + \eta \sum_{i=1}^{n_{\mathrm{tr}}} s_i \, \Theta(x, x_i), \tag{33}$$

so that after $T$ steps (or under gradient flow for time $\tau$) we obtain

$$f_T(x) \approx f_0(x) + c_T \sum_{i=1}^{n_{\mathrm{tr}}} s_i \, \Theta(x, x_i), \tag{34}$$

where $c_T$ is a (learning-rate / time-integrated) scalar.

Writing the signed sum as an empirical signed measure gives

$$\sum_{i=1}^{n_{\mathrm{tr}}} s_i \, \Theta(x, x_i) = \int_{\mathbb{Z}} \Theta(x, z) \, d(\hat{p} - \hat{q})(z), \tag{35}$$

revealing that, up to an overall scalar factor and the initialization bias $f_0$, training on (9) produces the *NTK witness function*:

$$f_T(\cdot) \approx f_0(\cdot) + c_T \int_{\mathbb{Z}} \Theta(\cdot, z) \, d(\hat{p} - \hat{q})(z). \tag{36}$$

Define the centered function $g(x) := f_T(x) - f_0(x)$. Under the NTK linearization, gradient flow yields a linear ODE for $g$ whose finite-time solution is the kernel expansion

$$g(x) = \sum_{i=1}^{n_{\mathrm{tr}}} \alpha_i(T) \, \Theta(x, x_i), \qquad \alpha_i(T) = c_T \, s_i, \tag{37}$$

where $c_T := \int_0^T \eta(t) \, dt$ collects the learning-rate schedule.

### A.3 PROOF OF LEMMA 3

Let $g(x)$ ignore the scalar $c_T$ (or absorb it into the kernel; it cancels in normalized comparisons). Evaluating $g$ on any finite set $X = \{x_k\}_{k=1}^{|X|}$ and averaging gives

$$\frac{1}{|X|} \sum_{x \in X} g(x) = \frac{1}{|X|} \sum_{x \in X} \int_{\mathbb{Z}} \Theta(x, z) \, d(\hat{p} - \hat{q})(z)$$
$$= \left\langle \frac{1}{|X|} \sum_{x \in X} \phi_\Theta(x), \, \int_{\mathbb{Z}} \phi_\Theta(z) \, d(\hat{p} - \hat{q})(z) \right\rangle_{\mathcal{H}_\Theta}. \tag{38}$$

where $\phi_\Theta$ is the (possibly implicit) feature map associated with the NTK $\Theta$ and $\mathcal{H}_\Theta$ its RKHS. Similarly for another set $B$:

$$\frac{1}{|Y|} \sum_{y \in Y} g(y) = \left\langle \frac{1}{|Y|} \sum_{y \in Y} \phi_\Theta(y), \, \int_{\mathbb{Z}} \phi_\Theta(z) \, d(\hat{p} - \hat{q})(z) \right\rangle_{\mathcal{H}_\Theta}. \tag{39}$$

Taking the difference,

$$\frac{1}{|X|} \sum_{x \in X} g(x) - \frac{1}{|Y|} \sum_{y \in Y} g(y)$$
$$= \left\langle \frac{1}{|X|} \sum_{x \in X} \phi_\Theta(x) - \frac{1}{|Y|} \sum_{y \in Y} \phi_\Theta(y), \, \int_{\mathbb{Z}} \phi_\Theta(z) \, d(\hat{p} - \hat{q})(z) \right\rangle_{\mathcal{H}_\Theta}. \tag{40}$$

The signed integral on the right becomes exactly the difference of their empirical mean embeddings, so that (40) reduces (up to the scalar $c_T$) to the squared MMD in the NTK RKHS:

$$\frac{1}{|X|} \sum_{x \in X} g(x) - \frac{1}{|Y|} \sum_{y \in Y} g(y) \;\propto\; \iint_{\mathbb{Z} \times \mathbb{Z}} \Theta(u,v)\, d(\hat{p}-\hat{q})(u)\, d(\hat{p}-\hat{q})(v) = \widehat{\mathrm{MMD}}_{\Theta}^2(\hat{p}, \hat{q}). \quad (41)$$

Thus, optimizing the parametric witness loss in the NTK (lazy training) regime yields, in function space, an estimator proportional to the empirical MMD with kernel given by the network's NTK.

## B  ADDITIONAL THEORETICAL RESULTS

### B.1  METRIC PROPERTY OF NTK–MMD

**Proposition B.1** (Metric property of NTK–MMD)**.** *Let $Z$ be a compact subset of $\mathbb{R}^d$ and let $\Theta : Z \times Z \to \mathbb{R}$ be a bounded, continuous, positive definite kernel. Denote by $\mathcal{H}_{\Theta}$ the associated RKHS and by*

$$\mathrm{MMD}_{\Theta}(P,Q) \;:=\; \big\| \mu_P - \mu_Q \big\|_{\mathcal{H}_{\Theta}} \quad for \quad \mu_P = \mathbb{E}_{x \sim P}[\phi(x)],\ \mu_Q = \mathbb{E}_{x \sim Q}[\phi(x)]$$

*the kernel MMD induced by $\Theta$ for probability measures $P, Q$ on $Z$. Then:*

1. $\mathrm{MMD}_{\Theta}(P,Q) \geq 0$ *for all $P, Q$, and $\mathrm{MMD}_{\Theta}(P,Q) = 0$ whenever $\mu_P = \mu_Q$;*

2. *if $\Theta$ is characteristic on $\mathcal{P}(Z)$, i.e., the mean embedding $P \mapsto \mu_P$ is injective, then $\mathrm{MMD}_{\Theta}$ defines a metric on $\mathcal{P}(Z)$:*

$$\mathrm{MMD}_{\Theta}(P,Q) = 0 \quad \Longleftrightarrow \quad P = Q.$$

*In particular, when $\Theta$ is universal (and hence characteristic) on the compact feature domain $Z$ considered in our NTK setting, the induced $\mathrm{MMD}_{\Theta}$ is a proper discrepancy metric.*

*Proof sketch.* Nonnegativity and the fact that $\mathrm{MMD}_{\Theta}(P,Q) = 0$ whenever $\mu_P = \mu_Q$ follow directly from the definition of the RKHS norm. When $\Theta$ is characteristic, the kernel mean embedding $P \mapsto \mu_P$ is injective on $\mathcal{P}(Z)$, so $\mu_P = \mu_Q$ implies $P = Q$, giving the "only if" direction. For bounded, continuous and universal kernels on compact domains, characteristicness is a standard consequence of universality; see, e.g., results of Vangeepuram (2010) and subsequent work on kernel mean embeddings. Combining these facts yields the metric property. $\square$

### B.2  TARGET-RISK BOUND AND NTK-BASED WEIGHTS

**Proposition B.2** (Target-risk bound and first-order optimality of NTK weights)**.** *Let $\Theta$ be a positive definite kernel on $Z$ with RKHS $\mathcal{H}_{\Theta}$. Suppose that for each label $y$, the loss $\ell(\cdot, y) : \mathbb{R} \to \mathbb{R}_+$ is $L$-Lipschitz with respect to the RKHS-induced pseudo-metric, in the sense that for any $f, g \in \mathcal{H}_{\Theta}$,*

$$\big| \ell(f(x), y) - \ell(g(x), y) \big| \;\leq\; L \big\| f - g \big\|_{\mathcal{H}_{\Theta}} \quad for\ all\ x \in Z.$$

*Assume $f \in \mathcal{H}_{\Theta}$ satisfies $\|f\|_{\mathcal{H}_{\Theta}} \leq B$. Let $P_S$ and $P_T$ denote the source and target distributions on $(x, y)$, and let $w : Z \to \mathbb{R}_+$ be any nonnegative weight function with $\mathbb{E}_{(x,y) \sim P_S}[w(x)] = 1$. Then the target risk*

$$R_T(f) \;:=\; \mathbb{E}_{(x,y) \sim P_T}\big[ \ell(f(x), y) \big]$$

*satisfies the bound*

$$R_T(f) \;\leq\; \mathbb{E}_{(x,y) \sim P_S}\big[ w(x)\, \ell(f(x), y) \big] \;+\; C\, \mathrm{MMD}_{\Theta}(P_S, P_T), \quad (42)$$

*for a constant $C$ depending only on $L$ and $B$ (e.g., $C \leq LB$ up to a universal factor).*

*Moreover, consider a gradual path $\{P_t\}_{t \in [0,1]}$ between $P_S$ and $P_T$ with associated witness function $g_t \in \mathcal{H}_{\Theta}$ defined by the NTK-MMD between $P_t$ and $P_T$. Then, to first order in $t$, the choice of weights*

$$w_t^{\star}(x) \;\propto\; 1 + \frac{g_t(x)}{\big\| \mu_{P_T} - \mu_{P_t} \big\|_{\mathcal{H}_{\Theta}}}$$

*minimizes the leading-order term of $R_T(f)$ along the path, aligning the weighted source risk with the direction of the domain shift induced by the NTK witness.*

*Proof sketch.* For the bound (42), write

$$R_T(f) - \mathbb{E}_{P_S}[w(x)\,\ell(f(x), y)] = \mathbb{E}_{P_T}[\ell(f(x), y)] - \mathbb{E}_{P_S}[w(x)\,\ell(f(x), y)].$$

Interpreting the right-hand side as an expectation of a bounded-Lipschitz functional over the difference of distributions and using the Lipschitz assumption together with $\|f\|_{\mathcal{H}_\Theta} \leq B$ yields a standard kernel-based domain adaptation bound in terms of $\mathrm{MMD}_\Theta(P_S, P_T)$, with a constant depending only on $L$ and $B$. This is analogous to classical MMD-based generalization bounds, specialized to the NTK-induced RKHS.

For the first-order optimality claim, one considers the directional derivative of the target risk along the path $\{P_t\}$ and linearizes around $t = 0$. The Gâteaux derivative of the risk functional with respect to perturbations of $P_t$ can be expressed in terms of the witness $g_t$ in $\mathcal{H}_\Theta$. A first-order expansion of the weighted source risk shows that choosing $w_t$ proportional to $1 + g_t/\|\mu_{P_T} - \mu_{P_t}\|_{\mathcal{H}_\Theta}$ cancels the leading-order mismatch between $P_t$ and $P_T$ and thus minimizes the linear term in the Taylor expansion of $R_T(f)$ along the path. We omit the full variational derivation for brevity. $\square$

### B.3 Finite-Width Deviation of NTK–MMD

**Proposition B.3** (Finite-width deviation of NTK–MMD). *Let $\Theta_m$ denote the (empirical) NTK of a width-$m$ network and $\Theta_\infty$ its infinite-width limit. For any two distributions $P, Q$ on $Z$, let $\Delta_\mu := \mu_P - \mu_Q$ denote the difference of their mean embeddings in the NTK feature space. Then the squared NTK–MMD satisfies*

$$\left| \mathrm{MMD}^2_{\Theta_m}(P, Q) - \mathrm{MMD}^2_{\Theta_\infty}(P, Q) \right| \leq \left\| \Theta_m - \Theta_\infty \right\|_{\mathrm{op}} \left\| \Delta_\mu \right\|_2^2, \tag{43}$$

*where $\| \cdot \|_{\mathrm{op}}$ denotes the operator norm.*

*Moreover, under standard wide-network assumptions for NTK convergence (e.g., Jacot et al. (2018); Lee et al. (2019); Arora et al. (2019b)), the operator norm deviation $\|\Theta_m - \Theta_\infty\|_{\mathrm{op}}$ concentrates at rate*

$$\left\| \Theta_m - \Theta_\infty \right\|_{\mathrm{op}} = \mathcal{O}_{\mathbb{P}}\big(m^{-1/2}\big),$$

*so that the finite-width approximation error in $\mathrm{MMD}^2_{\Theta_m}$ decays as $\mathcal{O}_{\mathbb{P}}(m^{-1/2})$.*

*Proof sketch.* By definition,

$$\mathrm{MMD}^2_{\Theta_m}(P, Q) = \langle \Delta_\mu, \Theta_m \Delta_\mu \rangle, \qquad \mathrm{MMD}^2_{\Theta_\infty}(P, Q) = \langle \Delta_\mu, \Theta_\infty \Delta_\mu \rangle.$$

Therefore

$$\left| \mathrm{MMD}^2_{\Theta_m}(P, Q) - \mathrm{MMD}^2_{\Theta_\infty}(P, Q) \right| = \left| \langle \Delta_\mu, (\Theta_m - \Theta_\infty)\Delta_\mu \rangle \right| \leq \left\| \Theta_m - \Theta_\infty \right\|_{\mathrm{op}} \left\| \Delta_\mu \right\|_2^2,$$

which proves (43). The stochastic rate $\|\Theta_m - \Theta_\infty\|_{\mathrm{op}} = \mathcal{O}_{\mathbb{P}}(m^{-1/2})$ is a direct consequence of concentration results for random feature/kernel matrices in the NTK regime: as the width $m$ grows, the empirical NTK converges to its deterministic infinite-width limit with fluctuations of order $m^{-1/2}$ in operator norm under standard initialization and scaling assumptions. We refer to the cited works for detailed proofs of this convergence. $\square$

# C  GRADUAL DOMAIN SHIFT EXPLORATION

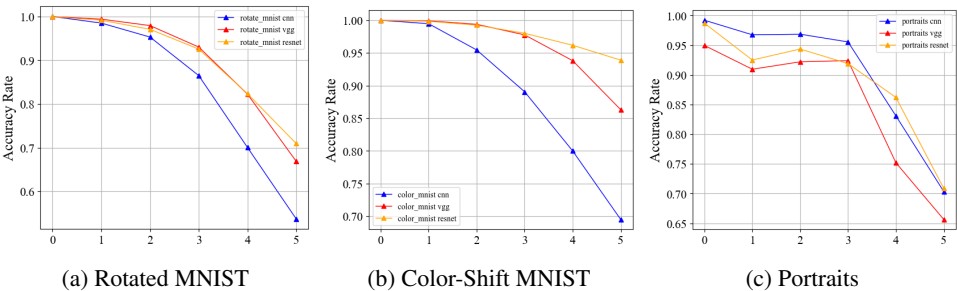

(a) Rotated MNIST  (b) Color-Shift MNIST  (c) Portraits

Figure 4: Classification accuracy under gradual domain shifts across three datasets. *Rotated MNIST*: constructed from MNIST, with the source domain comprising original digits and the target domain rotated by 45°. *Color-Shift MNIST*: source images are normalized to [0,1], while the target domain is shifted to [1,2], with intermediate domains generated via linear color interpolation. *Portraits*: real-world portrait images chronologically grouped into 9 temporal domains (1905–2013); the first and last domains are used as source and target, respectively.

To assess the robustness of different classifiers under gradual domain shifts, we evaluate three models—a small CNN, ResNet-18, and VGG11—across three benchmark datasets: Rotated MNIST, Color-Shift MNIST, and Portraits. As shown in Figure 4, all models are pretrained on the respective source domains and tested directly on target and intermediate domains without fine-tuning. This ensures a fair comparison by isolating the effects of distribution shift.

As domain shift severity increases, all classifiers experience accuracy degradation, with the small CNN consistently underperforming compared to deeper architectures. Notably, ResNet-18 and VGG11 demonstrate greater resilience, especially on Rotated and Color-Shift MNIST. These results highlight the varying sensitivity of model architectures to distributional shifts and emphasize the importance of evaluating robustness under gradual transitions.

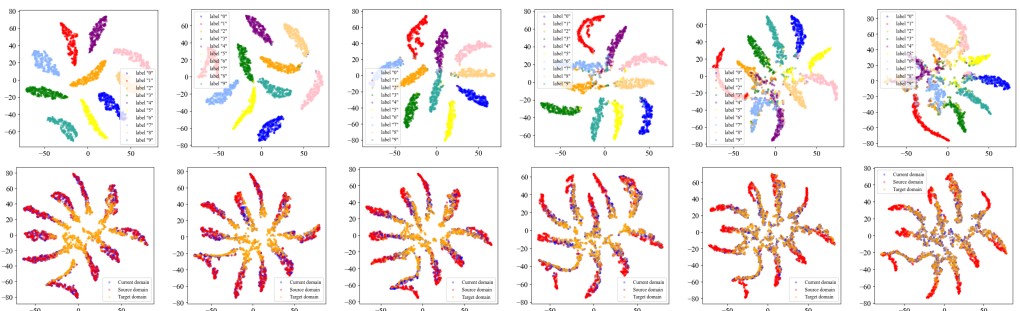

Figure 5: Visualization of feature representations across rotated domains using t-SNE. The top row shows the t-SNE visualizations of ResNet-18 encoded features for Rotated MNIST images across six domains with increasing rotation angles from 0° to 45°. Different colors represent different digit labels. The bottom row visualizes the feature distributions of the source domain (0°), target domain (45°), and the current domain at each adaptation step.

As shown in fig. 5, the feature representations extracted by ResNet-18 across different rotated domains reveal the impact of domain shift on classification boundaries. In the upper row, earlier domains exhibit well-separated classes, while later domains (closer to 45° rotation) show significant overlap between classes. The lower row illustrates the adaptation trajectory of the current domain, which initially aligns with the source domain and progressively moves toward the target distribution, demonstrating the effectiveness of the adaptation strategy in bridging the domain gap.

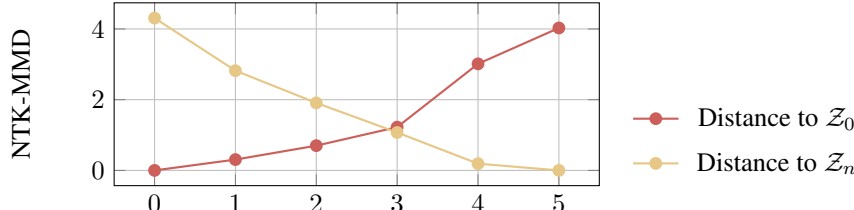

Figure 6: The figure shows the NTK-MMD distance between the current domain and the source domain $\mathcal{Z}_0$ (red), and between the current domain and the target domain $\mathcal{Z}_5$ (yellow), across six intermediate domains in Rotated MNIST (from 0° to 45°).

As visualized in fig. 6, the NTK-MMD distance between the current domain and the source domain $\mathcal{Z}_0$ increases with rotation angle, while the distance to the target domain $\mathcal{Z}_5$ decreases. This trend quantitatively reflects the smooth and continuous nature of domain shift in the Rotated MNIST setup, indicating that intermediate domains progressively depart from the source and approach the target in the feature space induced by ResNet-18.

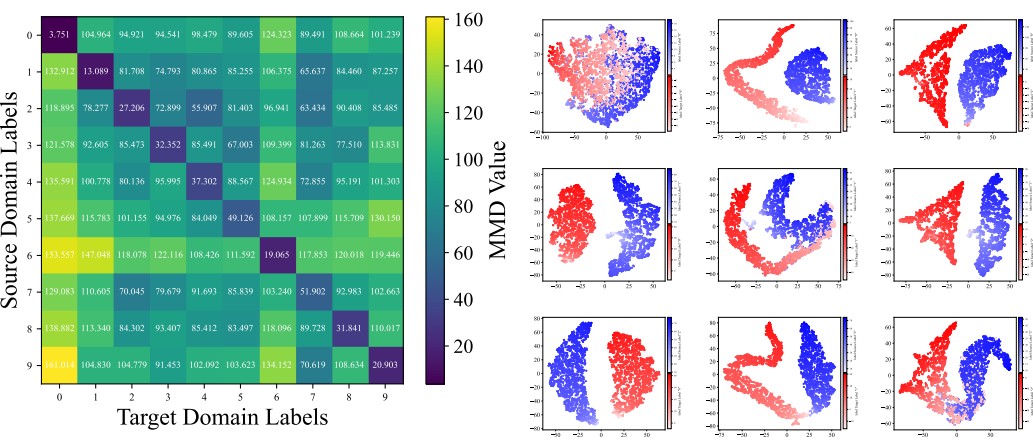

Figure 7: Cross-label NTK-MMD heatmap and t-SNE visualizations of label-wise feature alignment. The left panel shows a cross-label NTK-MMD heatmap between source and target domains on the Rotated MNIST dataset. Each cell denotes the MMD value between features of a specific source label (rows) and a target label (columns), computed using ResNet-18 representations. Higher values (brighter colors) indicate larger distributional differences. The right panel presents t-SNE visualizations of feature distributions for label pairs 0–3 in the source (blue) and target (red) domains. More intense colors indicate larger values in the NTK-reweighting function $w(z)$. These subplots illustrate the degree of alignment or mismatch corresponding to entries in the heatmap.

As shown in fig. 7, the left heatmap quantifies the distributional discrepancy between source and target features across all label combinations using NTK-MMD. Lower diagonal values suggest partial alignment of same-class features despite domain shift, while elevated off-diagonal values indicate potential cross-class confusion. The right-side t-SNE plots visualize the feature spaces of labels 0 to 3, providing qualitative support to the heatmap: well-separated red and blue clusters reflect higher MMD values, while overlapping clusters indicate better cross-domain consistency.

The heat-map intensities serve as a direct, visual proxy for the per-class discrepancy term $|g(z)|$ that appears in our NTK-reweighting acquisition score $a(z) = |g(z)|^\alpha / (\sqrt{\Theta(z,z)} + \epsilon)$ (see Sec. 3.4). Brighter blocks lie at the upper-left and lower-right corners of the grid, indicating source–target label pairs whose ResNet-18 features differ most strongly under the NTK-MMD. Because the denominator $\sqrt{\Theta(z,z)} + \epsilon$ merely normalises by the local kernel scale, these cells correspond to samples with the largest numerators—and hence the highest acquisition values when $\alpha = 1$. In practical terms, the figure tells us that digits such as "0" and "9" occupy feature regions that are both spa-

tially distant and poorly aligned across domains; prioritising them (i.e., assigning larger mini-batch weights during adaptation) accelerates the reduction of overall distributional mismatch, a pattern that is mirrored by the widely separated red–blue clusters in the accompanying t-SNE plots.

# D    GRADUAL DOMAIN ANALYSIS ON CIFAR-10-C AND CIFAR-100-C BENCHMARKS

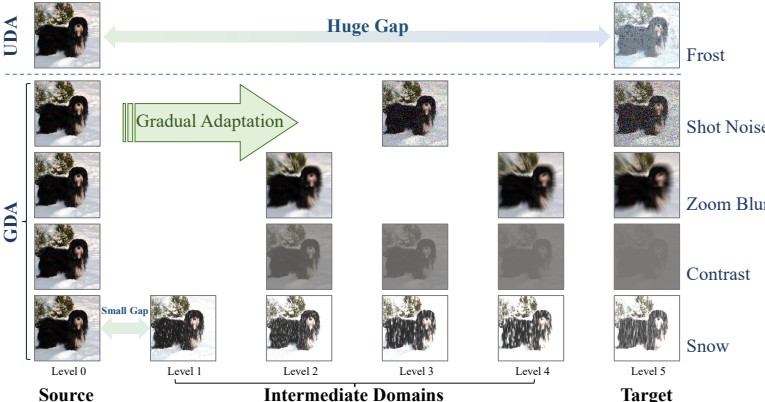

Figure 8: Step-wise curriculum of Gradual Domain Adaptation (GDA) on CIFAR-10-C / CIFAR-100-C. The corruption continuum is discretized into six severity levels (Level $0 \rightarrow 5$). A conventional unsupervised domain adaptation (UDA) model must traverse a *huge gap*, jumping directly from the pristine source images (Level 0) to the most severe target corruption (Level 5, here *Frost*). In contrast, GDA selects one or more *small-gap* intermediate domains (Levels 1–4) drawn from various corruption types (*Shot Noise*, *Zoom Blur*, *Contrast*, and *Snow* are illustrated) to form a smooth adaptation path that incrementally exposes the model to increasing difficulty.

In fig. 8 we visualize how GDA restructures the domain-shift challenge posed by CIFAR-C corruptions. Each benchmark image can be rendered at six predefined severity levels, which we treat as distinct domains. While classical UDA attempts to align the clean source distribution with the harshest target corruption in a single step, GDA inserts intermediate stops along this trajectory. For a given corruption type (e.g., *Zoom Blur*), the model may first adapt from Level 0 to Level 2, then Level 4, before finally reaching Level 5. By decomposing the shift into a sequence of milder transitions, GDA prevents abrupt feature drift, yields steadier convergence, and ultimately delivers superior robustness under severe corruptions.

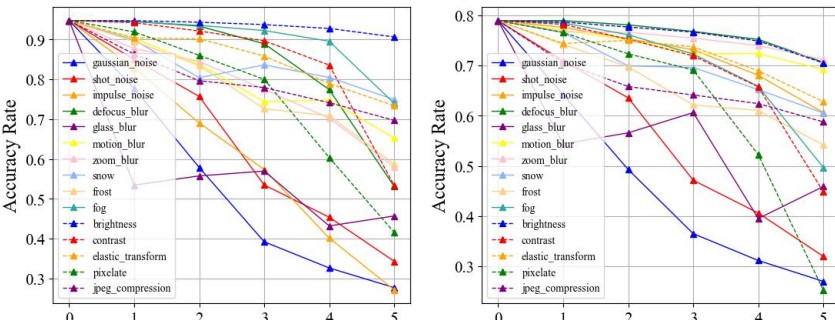

Figure 9: Classification accuracy under different corruption types and severities.Left: Accuracy of a WideResNet-28 classifier on CIFAR-10-C across 15 corruption types and 5 severity levels.Right: Accuracy of a ResNeXt-29 classifier on CIFAR-100-C under the same corruption conditions. Each line corresponds to a specific corruption type, and the x-axis represents increasing corruption severity from 0 to 5.

As shown in fig. 9, the classification performance of WideResNet-28 on CIFAR-10-C and ResNeXt-29 on CIFAR-100-C degrades progressively as the corruption severity increases across all 15 corruption types. This consistent and controllable degradation illustrates the suitability of CIFAR-10-C and CIFAR-100-C as benchmark datasets for studying gradual domain adaptation. Meanwhile, the varying rates of accuracy drop across corruption types highlight the inherent challenges posed by different domain shifts, underscoring the need for adaptive models that can robustly handle diverse and progressively shifting input distributions.

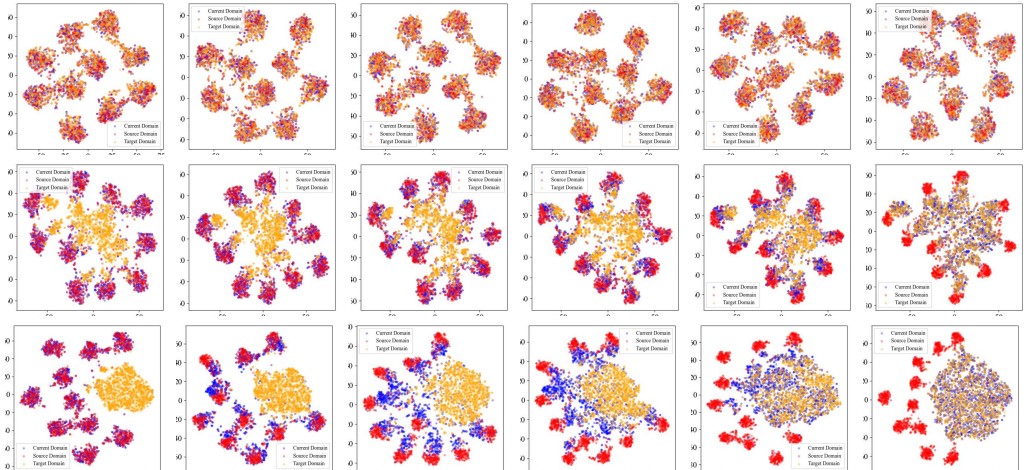

Figure 10: Progressive t-SNE visualizations of corrupted domains in CIFAR-10-C. Each row shows the feature distributions of six corruption severity levels (from top to bottom) for a specific corruption type in CIFAR-10-C: *brightness*, *contrast*, and *impulse noise*, respectively. Features are extracted using a WideResNet-28 encoder, and visualized using t-SNE. Red, blue, and yellow colors represent source, current, and target domains, respectively.

As shown in fig. 10, the progression of domain features under increasing corruption severity varies across corruption types. For the brightness corruption (top row), feature clusters remain well separated even under high distortion, suggesting robustness to this corruption, as also reflected in fig. 9. In contrast, the contrast and impulse noise rows (middle and bottom) show increasingly entangled feature distributions, especially at higher severity levels, indicating stronger domain shifts and degraded class separability.

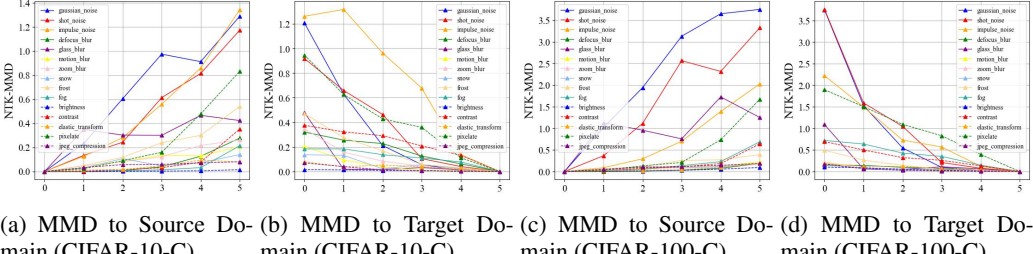

(a) MMD to Source Domain (CIFAR-10-C)  (b) MMD to Target Domain (CIFAR-10-C)  (c) MMD to Source Domain (CIFAR-100-C)  (d) MMD to Target Domain (CIFAR-100-C)

Figure 11: NTK-MMD distances to source and target domains across corruption severities. Each subplot shows the NTK-MMD distance between features of progressively corrupted data and the clean source domain or the most severely corrupted target domain for CIFAR-10-C (left, WideResNet-28) and CIFAR-100-C (right, ResNeXt-29). Results are computed over 15 corruption types and 5 severity levels.

As illustrated in fig. 11, the NTK-MMD distances between the current domain and the source domain increase with corruption severity, while the distances to the target domain (most severely corrupted) decrease accordingly. This consistent trend is observed on both CIFAR-10-C and CIFAR-100-C,

demonstrating a smooth and gradual shift in domain distributions. These characteristics confirm the suitability of CIFAR-C datasets as benchmarks for gradual domain adaptation, providing controllable and interpretable intermediate domains for evaluation.

# E MORE EXPERIMENTAL RESULTS

Table 8: Classification correctness (%) on the highest damage severity level 5 after gradual adaptation of CIFAR-10 to CIFAR-10-C with progressively higher damage. For all method results evaluated on the same WideResNet-28 that has been trained on the source domain. We report the average performance of our method over 5 runs.

| Method | # Given Domains | gaussian | shot | impulse | defocus | glass | motion | zoom | snow | frost | fog | brightness | contrast | elastic | pixelate | jpeg | Mean |
|---|---|---|---|---|---|---|---|---|---|---|---|---|---|---|---|---|---|
| Source | - | 27.7 | 34.3 | 27.1 | 53.1 | 45.7 | 65.2 | 58.0 | 74.9 | 58.7 | 74.0 | 90.7 | 53.3 | 73.4 | 41.5 | 69.7 | 56.5 |
| GST | 2 | 37.0 | 40.9 | 33.8 | 57.5 | 46.3 | 67.1 | 58.8 | 76.7 | 60.6 | 76.9 | 91.9 | 60.6 | 74.1 | 44.8 | 70.5 | 59.8 |
| | 3 | 38.3 | 44.0 | 39.1 | 73.8 | 46.5 | 70.1 | 68.1 | 80.2 | 65.1 | 79.4 | 92.6 | 80.4 | 71.5 | 46.4 | 71.2 | 64.5 |
| | 4 | 41.4 | 48.7 | 41.5 | 76.2 | 48.2 | 76.8 | 75.7 | 80.0 | 71.8 | 81.5 | 93.0 | 85.4 | 75.0 | 51.9 | 72.1 | 68.0 |
| | 5 | 42.5 | 50.3 | 45.2 | 79.2 | 49.1 | 78.0 | 78.5 | 80.4 | 72.2 | 82.0 | 93.1 | 86.1 | 75.0 | 54.1 | 73.1 | 69.2 |
| | 6 | 50.0 | 56.1 | 49.7 | 79.4 | 48.8 | 82.8 | 83.3 | 82.5 | 75.7 | 82.5 | 93.1 | 86.8 | 75.1 | 60.1 | 73.4 | 71.9 |
| GOAT | 2 | 27.7 | 34.3 | 27.1 | 53.1 | 45.8 | 65.3 | 58.1 | 74.9 | 58.7 | 74.0 | 90.7 | 53.4 | 73.5 | 41.6 | 69.8 | 56.5 |
| | 3 | 27.5 | 34.3 | 27.0 | 53.2 | 45.8 | 65.4 | 58.2 | 74.9 | 58.8 | 74.0 | 90.7 | 53.5 | 73.4 | 41.7 | 69.8 | 56.5 |
| | 4 | 27.4 | 34.2 | 26.9 | 53.2 | 45.6 | 58.2 | 65.4 | 75.0 | 59.0 | 74.0 | 90.6 | 53.4 | 73.6 | 41.8 | 69.8 | 56.5 |
| | 5 | 27.3 | 34.2 | 26.9 | 53.3 | 45.6 | 65.5 | 58.4 | 75.1 | 59.0 | 74.0 | 90.7 | 53.4 | 73.6 | 41.8 | 69.8 | 56.6 |
| | 6 | 27.3 | 34.3 | 27.0 | 53.3 | 45.5 | 65.7 | 58.5 | 75.1 | 59.0 | 74.0 | 90.7 | 53.4 | 73.6 | 41.9 | 69.8 | 56.6 |
| GradNTK (ours) | 2 | 65.5 | 67.4 | 60.6 | 81.7 | 64.2 | 83.4 | 83.8 | 84.2 | 82.5 | 85.3 | 94.4 | 86.0 | 77.8 | 72.0 | 74.4 | 77.5 |
| | 3 | 67.4 | 69.2 | 62.3 | 82.1 | 66.8 | 84.6 | 84.6 | 85.7 | 83.2 | 86.4 | 95.7 | 86.4 | 78.2 | 73.5 | 75.8 | 78.8 |
| | 4 | 68.8 | 71.7 | 63.7 | 83.6 | 68.8 | 86.0 | 86.3 | 87.2 | 84.9 | 87.7 | 96.6 | 87.8 | 79.3 | 76.5 | 77.4 | 80.4 |
| | 5 | 69.8 | 72.8 | 66.0 | 84.7 | 70.3 | 87.4 | 87.9 | 88.1 | 86.2 | 88.8 | 97.0 | 88.6 | 79.7 | 78.9 | 78.7 | 81.7 |
| | 6 | **70.7** | **73.6** | **67.0** | **85.6** | **71.6** | **88.2** | **88.9** | **88.9** | **87.0** | **89.4** | **97.4** | **89.3** | **80.3** | **81.4** | **79.9** | **82.6** |

**CIFAR-10-C Adaptation** As shown in table 8, we gradually increase the corruption level across up to six. As shown in Table 2, GradNTK steadily improves robustness: the mean accuracy at highest severity rises from 77.3% (2 domains) to 81.6% (6 domains), surpassing GST by over 9 points in the 6-domain setting. GOAT fails to leverage additional domains (flat at 56.6%), while GST only achieves 71.9% at six steps. This demonstrates GradNTK 's ability to exploit gradual shifts even under severe noise.

Table 9: Classification correctness (%) on the highest damage severity level 5 after gradual adaptation of CIFAR-100 to CIFAR-100-C with progressively higher damage. For all method results evaluated on the same ResNeXt-29 that has been trained on the source domain. We report the average performance of our method over 5 runs.

| Method | # Given Domains | gaussian | shot | impulse | defocus | glass | motion | zoom | snow | frost | fog | brightness | contrast | elastic | pixelate | jpeg | Mean |
|---|---|---|---|---|---|---|---|---|---|---|---|---|---|---|---|---|---|
| Source | - | 27.0 | 12.0 | 60.6 | 70.7 | 45.9 | 69.2 | 71.2 | 60.5 | 54.2 | 49.7 | 70.5 | 44.9 | 62.8 | 25.3 | 58.8 | 53.6 |
| GST | 2 | 49.4 | 52.3 | 62.4 | 72.4 | 51.9 | 70.4 | 72.6 | 64.7 | 63.0 | 56.5 | 73.7 | 61.0 | 64.0 | 50.2 | 60.0 | 61.6 |
| | 3 | 48.3 | 50.4 | 62.8 | 74.2 | 54.7 | 71.8 | 74.4 | 66.6 | 63.8 | 57.4 | 74.9 | 64.0 | 64.1 | 49.4 | 61.1 | 62.5 |
| | 4 | 47.7 | 49.5 | 64.2 | 75.6 | 56.1 | 73.0 | 75.7 | 67.3 | 65.6 | 59.1 | 76.5 | 67.2 | 65.5 | 50.3 | 61.9 | 63.7 |
| | 5 | 50.0 | 52.0 | 66.2 | 76.6 | 57.9 | 73.8 | 76.2 | 68.1 | 66.8 | 60.3 | 77.2 | 69.1 | 66.5 | 53.7 | 63.4 | 65.2 |
| | 6 | 50.2 | 53.3 | 67.7 | 77.5 | 58.6 | 75.0 | 76.7 | 69.7 | 67.8 | 61.9 | 77.9 | 73.0 | 66.9 | 59.2 | 64.2 | 66.7 |
| GOAT | 2 | 27.1 | 32.0 | 60.8 | 70.9 | 45.9 | 69.3 | 71.3 | 60.6 | 54.2 | 49.9 | 70.4 | 44.9 | 63.0 | 25.0 | 58.8 | 53.6 |
| | 3 | 26.8 | 31.9 | 60.7 | 71.0 | 45.9 | 69.6 | 71.3 | 60.6 | 54.2 | 49.8 | 70.4 | 45.1 | 63.1 | 25.2 | 58.7 | 53.6 |
| | 4 | 26.8 | 32.0 | 60.9 | 71.1 | 45.9 | 69.6 | 71.2 | 60.5 | 54.4 | 50.0 | 70.5 | 45.5 | 63.0 | 25.2 | 58.8 | 53.7 |
| | 5 | 26.3 | 32.1 | 60.9 | 71.1 | 46.0 | 69.7 | 71.3 | 60.6 | 54.3 | 50.0 | 70.6 | 45.9 | 63.2 | 25.2 | 58.8 | 53.7 |
| | 6 | 26.6 | 32.1 | 60.9 | 71.3 | 46.2 | 69.8 | 71.2 | 60.7 | 54.3 | 50.0 | 70.6 | 46.3 | 63.2 | 25.7 | 58.8 | 53.8 |
| GradNTK (ours) | 2 | 55.2 | 58.2 | 66.2 | 77.1 | 61.2 | 74.5 | 76.9 | 70.5 | 69.8 | 62.6 | 79.8 | 68.9 | 68.7 | 61.0 | 63.8 | 67.6 |
| | 3 | 57.6 | 61.7 | 69.9 | 80.3 | 63.9 | 77.4 | 79.8 | 73.5 | 72.3 | 65.2 | 83.6 | 71.0 | 71.0 | 64.3 | 66.3 | 70.5 |
| | 4 | 59.0 | 62.9 | 72.6 | 83.3 | 65.6 | 80.0 | 82.3 | 75.7 | 74.3 | 66.9 | 86.9 | 73.0 | 72.7 | 66.5 | 68.0 | 72.6 |
| | 5 | 60.0 | 64.3 | 75.1 | 85.8 | 67.2 | 81.9 | 84.5 | 77.7 | 76.0 | 68.8 | 89.9 | 74.9 | 74.2 | 68.7 | 69.6 | 74.6 |
| | 6 | **61.0** | **65.7** | **76.8** | **88.4** | **68.6** | **83.9** | **86.5** | **80.0** | **78.0** | **70.5** | **92.5** | **76.6** | **75.9** | **70.9** | **71.3** | **76.4** |

**CIFAR-100-C adaptation.** As shown in table 9, on CIFAR-100-C with a ResNeXt-29 backbone, GradNTK again outperforms: Table 3 shows the mean accuracy climbing from 67.8% (2 domains) to 76.5% (6 domains), an 8.8 point improvement. GST peaks at 66.7%, and GOAT remains at the source baseline of 53.8%.

# F    FEATURE DISTRIBUTION VISUALIZATION

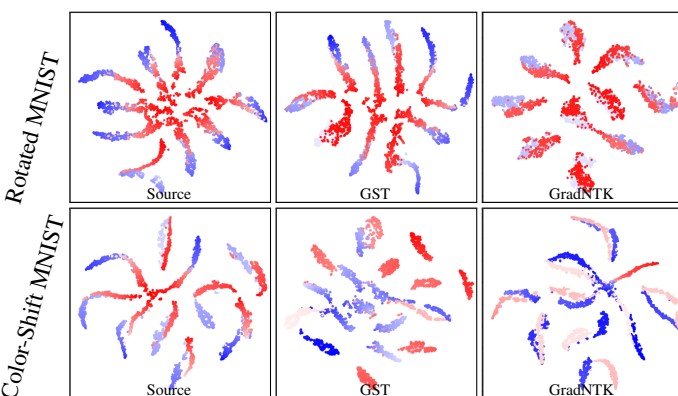

Figure 12: t-SNE visualizations of feature distributions on Rotated MNIST and Color-Shifted MNIST. Red and blue points represent features from the source and target domains, respectively. For each domain shift, feature representations are shown for three models: the baseline Source model (trained only on source data), the self-training baseline method (GST), and the proposed domain adaptation method (GradNTK). The visualizations highlight differences in domain alignment and representation learning under each approach.

fig. 12 illustrates the effectiveness of domain adaptation strategies in mitigating distribution shifts on two benchmark datasets: Rotated MNIST and Color-Shifted MNIST. Each row corresponds to a dataset, and each column compares feature distributions from three training paradigms. In both cases, the Source model exhibits distinct separation between source (red) and target (blue) features, indicating poor generalization to the shifted domains.

The GST model improves the spread of the target features, but without meaningful alignment to the source features. This supports the notion that GST primarily transfers knowledge from the source domain without enforcing alignment between feature spaces. As a result, semantic gaps remain, leading to potential misclassification and reduced domain robustness.

In contrast, the GradNTK model achieves significantly better feature overlap between domains, demonstrating clear alignment and cluster consistency. The blue and red points are co-located across most digit classes, suggesting that GradNTK successfully learns domain-invariant representations. This alignment is crucial for ensuring accurate and stable predictions in the target domain and validates GradNTK's design for effective unsupervised domain adaptation.

fig. 13 presents t-SNE visualizations that qualitatively assess the domain alignment capability of three different models under 15 corruption types from the CIFAR-10-C dataset. Each row corresponds to a specific corruption (e.g., *gaussian noise*, *motion blur*, *contrast*, etc.), while the columns display the feature distributions obtained by the Source-only model, the GST model without explicit domain adaptation, and the proposed GradNTK method.

The Source model shows a clear domain shift, with red (source) and blue (target) points forming largely disjoint clusters, indicating poor generalization to corrupted target domains. GST provides marginal improvements in feature mixing, but significant gaps still remain in many cases (e.g., *shot noise*, *pixelate*). In contrast, GradNTK demonstrates substantial improvement in feature alignment, with red and blue points forming tight, overlapping clusters in most visualizations. This suggests that GradNTK effectively bridges the domain gap and learns domain-invariant representations.

The consistency of this pattern across diverse corruption types highlights the robustness of GradNTK in handling severe distribution shifts. The visual evidence supports GradNTK's superior per-

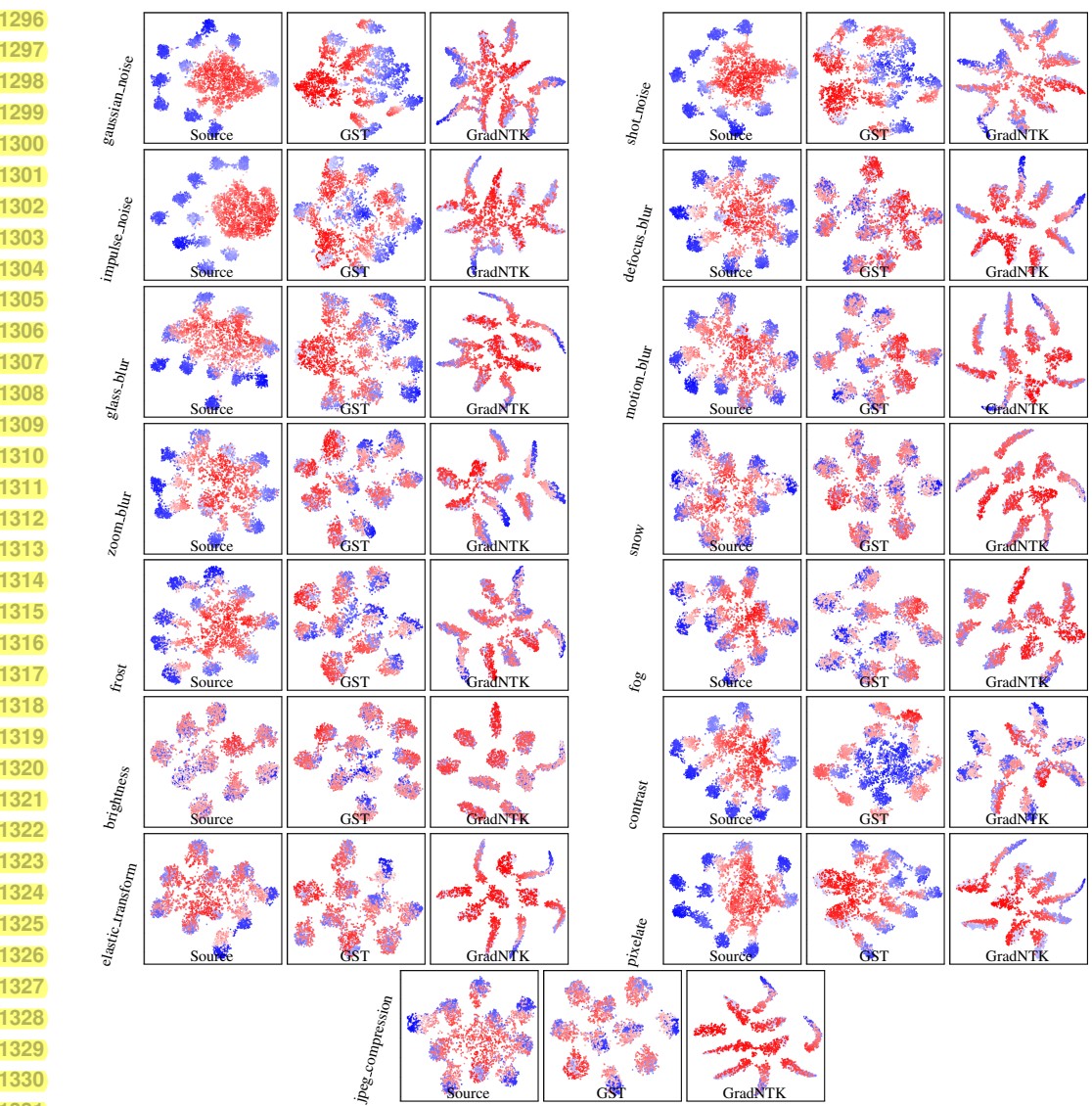

Figure 13: The t-SNE visualizations of learned features on the CIFAR-10-C dataset under various corruption types. Red points represent features from the source domain, while blue points denote features from the target domain. For each corruption, three methods are compared: the baseline encoder trained only on source data (Source), the encoder of baseline (GST), and our proposed domain adaptation approach (GradNTK). The visualizations illustrate the evolution of feature alignment across domains under increasing adaptation strategies. The heat-map intensities serve as a direct, visual proxy for the per-distribution discrepancy term $g(z)$

formance in unsupervised domain adaptation tasks, which is further corroborated by quantitative metrics in the main text.

## G   COMPUTATIONAL COST COMPARISON

The computational efficiency of our method is summarized in table 10, where we report both the running time and peak GPU memory usage across three benchmarks with four given domains. Compared to GST, GradNTK achieves a notable reduction in wall-clock time on all datasets. For example, on Rotated-MNIST, our method reduces training time from 164.9s to 95.6s (a 42% improvement), while on Color-Shift MNIST it reduces time from 164.0s to 104.8s. Similarly, for the Por-

Table 10: Running time and peak GPU memory usage (on an RTX 4090) for GradNTK, and GST across three benchmarks with 4 given domains and ResNet.

| Method | Rotated MNIST | Color-Shift MNIST | Portraits |
|---|---|---|---|
| GST | 164.9 s / 1218 MB | 164.0 s / 1218 MB | 5.6 s / 1350 MB |
| GradNTK | 95.6 s / 1604 MB | 104.8 s / 1604 MB | 4.5 s / 1842 MB |

traits benchmark, the runtime decreases from 5.6s to 4.5s. These consistent improvements indicate that GradNTK provides more efficient optimization dynamics during gradual adaptation.

In terms of memory footprint, GradNTK incurs a moderate increase in peak GPU usage relative to GST (e.g., 1604MB vs. 1218MB on MNIST variants, and 1842MB vs. 1350MB on Portraits). This overhead is expected due to the NTK-MMD regularization, yet it remains well within the capacity of standard modern GPUs. Overall, the results in table 10 demonstrate that GradNTK achieves substantially faster training while incurring only a modest increase in memory consumption, thereby offering a favorable trade-off between efficiency and performance.

## H  PSEUDOCODE

---

**Algorithm 1:** GradNTK training loop

---

**Input:** Source dataset $D_0$, target domains/stream $\{D_t\}$
**Hyperparams:** $\lambda_{\mathrm{MMD}} \geq 0$, $\lambda_{\mathrm{src}} \geq 0$, drop rate $q \in [0,1)$, temp $T > 0$, epochs $E$, batch size $b$
**Initialize:** Model $h(x; \psi, \varphi) = k(r(x; \psi); \varphi)$; optimizer on $(\psi, \varphi)$
**foreach** *target domain* $D_t$ **do**
    Build loaders $\mathcal{L}_s \leftarrow \mathrm{DataLoader}(D_0, b)$, $\mathcal{L}_t \leftarrow \mathrm{DataLoader}(D_t, b)$;
    **for** iter $= 1$ **to** $\max(|\mathcal{L}_s|, |\mathcal{L}_t|) \cdot E$ **do**
        Sample $(x_s, y_s) \sim \mathcal{L}_s$, $x_t \sim \mathcal{L}_t$;    $e_s \leftarrow r(x_s; \psi)$, $e_t \leftarrow r(x_t; \psi)$;
        $(\mathcal{L}_{\mathrm{MMD}}, g_s, g_t) \leftarrow \mathtt{MMDAndWitness}(e_s, e_t)$; $w_s \leftarrow \mathtt{SoftmaxT}(g_s, \tau)$,
        $w_t \leftarrow \mathtt{SoftmaxT}(g_t, \tau)$;
        $\ell_t \leftarrow k(e_t)$;    $\hat{y}_t \leftarrow \mathtt{argmax}(\ell_t)$, $\mathrm{conf} \leftarrow (\ell_t)_{(1)} - (\ell_t)_{(2)}$;
        $\alpha \leftarrow \mathtt{Quantile}(\mathrm{conf}, q)$;    $\mathcal{M} \leftarrow \{\mathrm{conf} \geq \alpha\}$;
        $L_{\mathrm{NR}} \leftarrow \mathrm{CE}\big(\ell_t[\mathcal{M}], \hat{y}_t[\mathcal{M}]; w_t[\mathcal{M}]\big) + \mathrm{CE}\big(k(e_s), y_s; w_s\big)$;
        $L \leftarrow L_{\mathrm{NR}} + \lambda_{\mathrm{MMD}}\mathcal{L}_{\mathrm{MMD}}$;
        Update $(\psi, \varphi)$ by one optimizer step on $L$;

---

## LLM USAGE DISCLOSURE

We used a large language model (LLM)—OpenAI ChatGPT (GPT-5)—accessed between July– September 2025. Below we describe its precise roles and our safeguards.

**Scope of use.**

1. *Writing & editing:* language polishing (grammar, clarity, concision), suggestions on paragraph flow, and occasional rephrasing of sentences in the Introduction, Method exposition, and Discussion.

2. *Structural support:* help drafting outlines for sections and checklists (e.g., what to report for datasets, ablations, and hyperparameters), and converting bulleted notes into readable prose that we then revised.

3. *LaTeX assistance:* resolving compilation errors (e.g., `pgfplots`/TikZ warnings), generating boilerplate for tables/figures/captions, and minor formatting macros (no mathematical content was introduced by the LLM).

4. *Code scaffolding:* small non-novel utilities such as argument parsers, experiment-launch bash snippets, logging wrappers, and plotting stubs. All core algorithmic code (losses, NTK/MMD components, training loop) was written and verified by the authors.

5. *Literature QA:* help recalling venue/year for already-known papers and checking citation style. We did **not** accept bibliographic details without verification; all citations were cross-checked against the original papers.

**Non-uses / boundaries.**

- The LLM did not originate the research idea, define the method, select experiments, tune hyperparameters, run evaluations, or choose which results to report.

- The LLM did not produce proofs or theoretical claims; any proof wording it helped with was limited to editorial clarity on author-written arguments.

- The LLM did not write Related Work content based on uncited or unverifiable sources.

