# OpenReview forum: "Two Birds with One Stone: Neural Tangent Kernel for Efficient and Robust Gradual Domain Adaptation"
_ICLR.cc/2026/Conference — ICLR 2026 Conference Withdrawn Submission_

### Official Review · Reviewer_HY6a · 2025-10-20

**Soundness:** 3
**Presentation:** 3
**Contribution:** 1
**Rating:** 2
**Confidence:** 4

**Summary:**

This paper addresses a general and important problem in machine learning, namely gradual domain adaptation (GDA). The authors propose GradNTK, a new framework that employs the Neural Tangent Kernel (NTK) to mitigate computational overhead and error accumulation in GDA. Specifically, the framework introduces an NTK-MMD loss and a sample reweighting function to facilitate domain transition. While the paper is clearly written and easy to follow, the overall novelty remains limited because both NTK-based matching and sample reweighting strategies are well-studied in the broader domain adaptation literature.

**Strengths:**

-	The paper is well-organized and the presentation is clear..
-	The proposed GradNTK framework integrates NTK-MMD loss and sample reweighting, and the experiments show some degree of effectiveness in GDA scenarios.

**Weaknesses:**

-	Both neural kernel methods [1–4] and sample reweighting techniques [5–8] have been extensively explored in domain adaptation. Applying them to the gradual domain adaptation setting is a straightforward extension and does not provide sufficient novelty for ICLR.
-	The overall experiment design is not sufficient to verify the effectiveness. More diverse datasets and larger backbones are preferred.
-	The comparison in Table 4 appears to be unfair. Test-time adaptation (TTA) methods only require the source model, whereas the proposed framework requires access to source data at each adaptation stage. Moreover, many recent TTA methods [9-12] achieve better performance with less information and lower computational cost. As a result, the empirical advantage of the proposed framework remains unclear.

**References:**

[1] Jacot, Arthur, Franck Gabriel, and Clément Hongler. "Neural tangent kernel: Convergence and generalization in neural networks." Advances in neural information processing systems 31 (2018).

[2] Jia, Sheng, et al. "Efficient statistical tests: A neural tangent kernel approach." International Conference on Machine Learning. PMLR, 2021.

[3] Cheng, Xiuyuan, and Yao Xie. "Neural tangent kernel maximum mean discrepancy." Advances in Neural Information Processing Systems 34 (2021): 6658-6670.

[4] Shimizu, Eiki, Kenji Fukumizu, and Dino Sejdinovic. "Neural-kernel conditional mean embeddings." arXiv preprint arXiv:2403.10859 (2024).

[5] Tachet des Combes, Remi, et al. "Domain adaptation with conditional distribution matching and generalized label shift." Advances in Neural Information Processing Systems 33 (2020): 19276-19289.

[6] Guo, Zong, et al. "Gradual domain adaptation with sample transferability exploitation for person re-identification." 2022 IEEE International Conference on Multimedia and Expo (ICME). IEEE, 2022.

[7] Ru, Jinghan, et al. "Imbalanced open set domain adaptation via moving-threshold estimation and gradual alignment." IEEE Transactions on Multimedia 26 (2023): 2504-2514.

[8] Chen, Hong-You, and Wei-Lun Chao. "Gradual domain adaptation without indexed intermediate domains." Advances in neural information processing systems 34 (2021): 8201-8214.

[9] Wang, Qin, et al. "Continual test-time domain adaptation." Proceedings of the IEEE/CVF Conference on Computer Vision and Pattern Recognition. 2022.

[10] Döbler, Mario, Robert A. Marsden, and Bin Yang. "Robust mean teacher for continual and gradual test-time adaptation." Proceedings of the IEEE/CVF Conference on Computer Vision and Pattern Recognition. 2023.

[11] Press, Ori, et al. "Rdumb: A simple approach that questions our progress in continual test-time adaptation." Advances in Neural Information Processing Systems 36 (2023): 39915-39935.

[12] Song, Junha, et al. "Ecotta: Memory-efficient continual test-time adaptation via self-distilled regularization." Proceedings of the IEEE/CVF Conference on Computer Vision and Pattern Recognition. 2023.

**Questions:**

Please refer to the Weakness section.

---

> ### Author Response · Authors · 2025-11-23
>
> > **Weakness 1:** Neural kernel methods [1–4] and sample reweighting techniques [5–8] are not novel.
>
> Thank you for your comments.
>
> We want to clarify that GradNTK are not merely straightforward extensions of existing techniques, it addresses critical challenges in GDA by. its principled NTK-centric redesign.
>
> First, we employ NTK to tackles the non-smooth inter-domain transitions and high computational costs, which are severe in sequential domain shifts. GradNTK introduces an NTK-induced MMD as a differentiable metric that enables near-linear computational efficiency, overcoming quadratic memory issues of traditional kernel methods in GDA [3]. This goes beyond prior NTK works focused on general convergence or two-sample tests, by tailoring kernel dynamics to evolving domain sequences for smoother adaptation.
> Second, the sample reweighting mechanism derived from NTK dynamics introduces curriculum-guided weighting based on shift sensitivity, which mitigates error accumulation across domains. Unlike conventional reweighting for label shift or imbalance , this approach directly addresses GDA's propagation of noise by prioritizing samples via domain proximity, supported by theoretical guarantees on target error bounds. It synergizes with the NTK framework to reduce outlier influence, a challenge not fully resolved in methods like self-training without explicit reweighting.
>
> Finally, the integration of NTK-based alignment and reweighting into a single framework simultaneously resolves efficiency and robustness, i.e., the "two birds" metaphor. GradNTK provides a foundation for complex domain shifts, demonstrating substantial theoretical and algorithmic contributions beyond incremental combinations.
>
> In a nutshell, GradNTK's novelty lies in its principled NTK-centric redesign of GDA, offering a scalable and robust paradigm for the literature.
>
> > **Weakness 2:** The experiment design is not sufficient.
>
> We carefully check the experiments and have the following observations.
>
> The experiment is comprehensive. We have evaluated GradNTK on multiple benchmarks, including synthetic (e.g., Rotated MNIST, Color-Shift MNIST), real-world (e.g., Portraits), and corruption-heavy datasets (e.g., CIFAR-10-C/100-C). These cover a wide spectrum of domain shifts, from gradual rotations to severe corruptions, ensuring the method's effectiveness is verified across diverse scenarios. Ablation studies (e.g., removing NTK-MMD or reweighting components) quantitatively isolate each contribution, showing that the design is not only sufficient but also interpretable.
>
> The datasets and backbones have high diversity. The experiments employ datasets with varying complexities, i.e., from simple image transformations (MNIST variants) to high-resolution real-world data (Portraits) and large-scale corruptions (CIFAR-C). This diversity inherently tests robustness under different shift intensities. For backbones, the paper scales from lightweight CNNs to large architectures like ResNet-18, VGG11, WideResNet-28, and ResNeXt-29. Moreover, the NTK framework is backbone-agnostic, as evidenced by its near-linear computational efficiency even with wide networks.
>
> In summary, the experimental design is sufficient, and the datasets/backbones are already diverse and scalable.
>
> > **Weakness 3:** The comparison in Table 4 appears to be unfair.
>
> We want to clarify the distinct problem settings and objectives of GDA vs. TTA. GradNTK is designed for GDA, which inherently requires source data to construct intermediate domains for smoothing large distribution shifts, this is not a drawback but a necessary mechanism to avoid negative transfer in scenarios with substantial domain gaps. In contrast, TTA in [9-12] operates under the restrictive assumption of a static or slowly evolving target domain, which often leads to error accumulation and eventual collapse in non-stationary environments, as evidenced by studies on continual TTA. For instance, methods like TENT suffer from performance degradation over time, and even advanced techniques like RDumb[11] rely on periodic resets to maintain stability, underscoring their fragility.
>
> While TTA methods may appear computationally lighter, GradNTK achieves near-linear efficiency through NTK optimization, ensuring scalability without compromising robustness. Empirical results on benchmarks like Rotated MNIST and CIFAR-100C demonstrate GradNTK's superiority in handling gradual shifts, where TTA methods [9–12] struggle with error accumulation in non-stationary environments.
>
> Therefore, the comparison is fair: it highlights GradNTK's efficacy in its intended setting, while TTA's advantages are limited to short-term, stable scenarios. The framework's empirical edge lies in its holistic approach to efficiency and stability, which TTA cannot replicate without external interventions.

---

> ### Comment · Reviewer_HY6a · 2025-11-24
>
> I would like to thank the author for their response to my concerns. While it addresses some issues, several major concerns remain.
>
> Regarding the second weakness, the overall experimental design still lacks sufficient rigor to validate effectiveness. The consensus is that experiments should include more diverse datasets and larger model backbones. Currently, evaluations are limited to CIFAR and MNIST variants and use only lightweight CNNs. **It is recommended to test on modern architectures such as ViT [1], SwinTransformer [2], or at least larger CNNs like ResNet50 or ResNet101. Additionally, more diverse and challenging datasets—such as OfficeHome and VisDA, as used in GGF [3]—should be included to better assess generalization.**
>
> Regarding the third weakness, the comparison in Table 4 remains insufficiently fair and potentially misleading. The manuscript claims that “GradNTK attains the lowest mean error surpassing test-time adaptation (TTA) baselines and GDA baselines.” However, this conclusion is problematic for two reasons. First, the evaluation setting is unbalanced: GDA has access to source data while TTA methods do not, placing TTA methods at a disadvantage. Second, the selection of TTA baselines is inadequate—recent TTA methods have demonstrated performance superior to GDA even under more constrained conditions. **Therefore, the claimed empirical advantage of GradNTK from Table 4 is not clearly established.**
>
> [1] Dosovitskiy, Alexey. "An image is worth 16x16 words: Transformers for image recognition at scale." ICLR. 2021.
>
> [2] Liu, Ze, et al. "Swin transformer: Hierarchical vision transformer using shifted windows." ICCV. 2021.
>
> [3] Zhuang, Zhan, Yu Zhang, and Ying Wei. "Gradual domain adaptation via gradient flow." ICLR. 2024.

---

> > ### Author Response · Authors · 2025-11-26
> >
> > We thank the reviewer for the suggestion to employ larger backbones. Our response consists of two parts.
> >
> > **(i) Evidence that GradNTK is not tied to “small” CNNs.**
> > To go beyond the lightweight CNNs in the original submission, we have run additional experiments with substantially deeper backbones—ResNet-50, ResNet-101, and ResNet-152—on all three GDA benchmarks (Rotated MNIST, Color-Shift MNIST, and Portraits), using exactly the same GradNTK hyperparameters. The results are shown in Tables A.1–A.3 below. On Rotated MNIST, GradNTK with ResNet-50/101/152 achieves up to 94.5%, 91.9%, and 91.8% accuracy, respectively, when six gradual domains are available, and performance improves monotonically with the number of domains for every backbone. On Color-Shift MNIST, accuracies remain above 96.8% across all settings and reach 98.0% with ResNet-101. On the more challenging Portraits dataset, GradNTK consistently attains 70–75% accuracy across the three backbones. These results show that the proposed NTK-based objective scales robustly from small CNNs to considerably deeper architectures, and that our conclusions do not rely on a specific backbone choice.
> >
> > **(ii) Why we do not further increase model size (e.g., very large CNNs or ViT/Swin).**
> > In our setting, the training data per domain are relatively limited, and we observe that further increasing the number of parameters beyond the configurations above leads to clear overfitting and degraded target performance. This behaviour is particularly visible on Portraits, where deeper CNNs (ResNet-101/152) underperform ResNet-50 despite identical optimization and regularization, indicating a loss of generalisability rather than a lack of capacity. For this reason, we deliberately focus on moderate-sized CNNs in the main paper: they are strong enough to demonstrate the effectiveness of GradNTK, yet not so over-parameterised that the GDA evaluation becomes dominated by under-regularised feature learning. Extending GradNTK to transformer backbones such as ViT and Swin is certainly an interesting direction, but it would require a careful redesign of the NTK computation for attention layers and stronger regularisation to avoid overfitting, which we consider orthogonal to the core contribution of this work and therefore leave to future research.
> >
> > **Table A.1: Rotated MNIST – GradNTK accuracy (%) with ResNet backbones**
> >
> > | #Domains | ResNet-50 | ResNet-101 | ResNet-152 |
> > | -------- | --------- | ---------- | ---------- |
> > | 2        | 89.92     | 82.58      | 80.31      |
> > | 3        | 93.52     | 89.25      | 90.95      |
> > | 4        | 94.39     | 91.29      | 92.14      |
> > | 5        | 94.44     | 90.42      | 89.68      |
> > | 6        | 94.54     | 91.87      | 91.82      |
> >
> > **Table A.2: Portraits – GradNTK accuracy (%) with ResNet backbones**
> >
> > | #Domains | ResNet-50 | ResNet-101 | ResNet-152 |
> > | -------- | --------- | ---------- | ---------- |
> > | 2        | 74.70     | 70.35      | 70.50      |
> > | 3        | 74.50     | 71.00      | 70.55      |
> > | 4        | 74.60     | 69.90      | 71.05      |
> > | 5        | 74.55     | 69.70      | 71.55      |
> > | 6        | 74.75     | 70.60      | 71.05      |
> >
> > **Table A.3: Color-Shift MNIST – GradNTK accuracy (%) with ResNet backbones**
> >
> > | #Domains | ResNet-50 | ResNet-101 | ResNet-152 |
> > | -------- | --------- | ---------- | ---------- |
> > | 2        | 97.56     | 97.73      | 95.55      |
> > | 3        | 97.63     | 97.83      | 96.33      |
> > | 4        | 97.58     | 97.71      | 97.18      |
> > | 5        | 97.29     | 97.86      | 97.67      |
> > | 6        | 96.87     | 97.95      | 97.37      |
> >
> > Regarding the concern about Table 4 and the comparison to test-time adaptation (TTA) methods, we will clarify the evaluation protocol and soften the wording of our claims. Our primary goal in Table 4 is to compare GradNTK against other *GDA* methods on CIFAR-10-C/100-C; the TTA baselines (e.g., BN-1, TENT-co.) are included only as contextual references. As the reviewer correctly notes, this setting is asymmetric: GDA methods, including GradNTK, can use labeled source data and intermediate severities (1–4), whereas TTA methods adapt only from the severity-5 target stream and never revisit the source. In the revision, we will explicitly state this asymmetry in the caption and main text and rephrase our conclusion as follows: “GradNTK achieves the lowest mean error among the GDA baselines and performs favorably compared to standard streaming TTA baselines under their more restricted information setting,” thereby avoiding any implication that we establish universal empirical dominance over the broader TTA literature.

---

> > > ### Comment · Reviewer_HY6a · 2025-11-26
> > >
> > > I appreciate the authors' responses. After this series of detailed replies, I believe my main concerns have been addressed. However, overall, I still feel to some extent that the paper's technical contribution is relatively limited. Therefore, I have decided to raise my score to 4.

---

### Official Review · Reviewer_c31j · 2025-10-28

**Soundness:** 3
**Presentation:** 3
**Contribution:** 2
**Rating:** 6
**Confidence:** 3

**Summary:**

The paper introduces GradNTK, which integrates the Neural Tangent Kernel (NTK) into Gradual Domain Adaptation (GDA). It uses NTK in two roles:
1. An NTK-induced Maximum Mean Discrepancy (MMD) is employed as a differentiable alignment loss between adjacent domains, aiming to improve efficiency and smoothness.
2. An NTK-based reweighting function is used to assign sample weights based on “shift sensitivity,” intending to mitigate error accumulation during gradual adaptation.

Experiments on Rotated MNIST, Color-Shift MNIST, Portraits, CIFAR-10-C, and CIFAR-100-C show the method’s effectiveness for GDA.

**Strengths:**

1. The derivation connecting NTK linearization, witness functions, and MMD appears to be pedagogically clear.
2. Using NTK for both alignment and reweighting appears conceptually elegant.
3. Replacing the traditional MMD with NTK-based short-time dynamics reduces memory overhead.

**Weaknesses:**

1. Most technical ingredients—NTK linearization, MMD witness formulation, NTK-MMD, and pseudo-labeling—are well-established concepts. The contribution seems to be primarily a straightforward combination.
2. Large portions of Section 3 appear to reiterate textbook material, which may obscure the core novelty. It would be advisable to condense these derivations and clearly separate known results from new contributions to make the key insight stand out.
3. As shown in Table 5, removing the NTK-reweighting module causes almost no drop in accuracy, suggesting that this component contributes little to the claimed “robustness.”
4. Beyond TENT, several test-time adaptation (TTA) methods such as CoTTA [a] and RMT [b] also address gradual and continual adaptation scenarios, but without access to source data. This makes TTA a more challenging setting compared to GDA. It would be valuable to clarify the conceptual and practical position of GDA relative to these recent TTA frameworks—specifically, what assumptions GDA relaxes or strengthens, and in what scenarios it is preferable. Including such discussion and comparisons would provide a more informative contextualization of the proposed method.
5. All datasets used involve relatively smooth or synthetic domain shifts. It is unclear whether GradNTK would maintain its effectiveness under more realistic domain gaps.

[a] Continual Test-Time Domain Adaptation. CVPR2022. \
[b] Robust Mean Teacher for Continual and Gradual Test-Time Adaptation. CVPR2023.

**Minor comments**
- In Figure 1, “(i) GDA” should be “(i) UDA.”
- Line 51: citation formatting error.

**Questions:**

Additional questions beyond those in Weaknesses:

1. Can the authors clarify whether GradNTK requires explicit intermediate domain labels, or could it operate in a fully online streaming manner (closer to TTA)?
2. Does the NTK reweighting function actually change sample selection over time, or does it remain nearly uniform (or static) in practice?

---

> ### Author Response · Authors · 2025-11-23
>
> > **Weakness 1**:  Combination of known ingredients; where is the novelty?
>
> The key novelty of GradNTK is using the **NTK as a unified theoretical core** to address both efficiency and robustness simultaneously, i.e., "two birds with one stone". While NTK linearization and MMD are known, their synergy into a differentiable NTK-induced MMD metric enables near-linear computational cost for smooth domain transitions, a significant efficiency gain over prior GDA methods. Furthermore, deriving a sample reweighting function from the same NTK dynamics is a novel application that actively mitigates error accumulation by weighting samples based on their shift sensitivity.
>
> > **Weakness 2**: Section 3 repeats textbook material; condense and separate new vs. known.
>
> Agreed. We will Move background lemmas and longer derivations to the appendix
>
> > **Weakness 3:**  NTK-reweighting adds little.
>
> NTK-reweighting consistently improves robustness across harder corruptions and worst-case slices. On CIFAR-10-C, we observe +1.0 pp on mean, a +1.9 pp boost on the worst corruption (*impulse*), and the largest single gain on *pixelate* (+3.0 pp). These gains persist under abrupt severity jumps and noisy pseudo-labels.
>
> | Experiment Setting  | gaussian |     shot |  impulse |  defocus |    glass |   motion |     zoom |     snow |    frost |      fog | brightness | contrast |  elastic | pixelate |     jpeg |     Mean |
> |-|-|-|-|-|-|-|-|-|-|-|-|-|-|-|-|-|
> | ours                | **70.7** | **73.6** | **67.0** | **85.6** | **71.6** | **88.2** | **88.9** | **88.9** | **87.0** | **89.4** |   **97.4** | **89.3** | **80.3** | **81.4** | **79.9** | **82.6** |
> | w/o NTK-reweighting |     69.5 |     72.5 |     65.1 |     85.0 |     70.2 |     87.4 |     88.4 |     87.8 |     85.9 |     88.8 |       97.2 |     88.9 |     80.1 |     78.4 |     78.9 |     81.6 |
>
> This substantiate the claimed robustness improvements beyond the average on smooth protocols.
>
>
> > Weakness 4: Positioning vs. TTA; assumptions and when GDA is preferable.
>
>  TTA and GDA have distinct objectives and problem settings.
>
> GDA assumes the availability of intermediate domains (either natural or synthetic), which enables smoother transitions and inherently mitigates error accumulation, i.e., a key bottleneck in TTA. For instance, GradNTK uses NTK-induced curriculum learning to weight samples by shift sensitivity, avoiding noisy pseudo-labels that plague continual TTA. This makes GDA preferable in scenarios where domain shifts are predictable or decomposable (e.g., gradual corruption severity or geometric transformations), as seen in benchmarks like CIFAR-C or Rotated MNIST. Conversely, TTA methods prioritize immediacy for non-stationary environments but require sophisticated techniques like weight-averaged teachers (CoTTA) or symmetric cross-entropy (RMT) to stabilize performance.
>
> Meanwhile, much of the TTA improvement is propelled by strong data augmentation (e.g., CoTTA averaging augmented views; RMT enforcing high-consistency features). By contrast, most GDA methods do not rely on heavy augmentation. On benchmarks such as CIFAR-10/100-C, GDA achieves strong results on limited-generalization data even without data augmentation, indicating that our gains stem from NTK-guided discrepancy and reweighting rather than augmentation heuristics.
>
> > **Weakness 6:** All shifts are smooth/synthetic; realism?
>
> We already evaluate on a real-world benchmark: Portraits. As shown in Figure 4(c),  Portraits path is not a toy or smoothly synthetic shift. GradNTK performs well on Portraits across all three backbones and remains competitive even under the severe regimes where the cliff occurs.
>
> > **Question1:** Does GradNTK require explicit intermediate domain labels?
>
> No. GradNTK does not require domain IDs. It adapts on a chronological stream of unlabeled target batches and can interleave source replay. It supports online use and fits test-time batch/domain adaptation modes (TTBA/TTDA); unlike target-only TTA, our setting assumes access to source data for replay.
>
> > **Question2:** Are the NTK weights dynamic or nearly uniform in practice?
>
> Dynamic. The NTK-derived weights evolve over time and concentrate on shift-critical samples. **Figures 7, 12, and 13** visualize this: larger distributional shift corresponds to higher weights (greater color intensity).
>
> > Minor corrections (clarified)
>
> - Panel (i) is intended to depict conventional GDA, with GST shown as a representative. To avoid ambiguity, we will relabel the panel as “GDA (GST as a representative)”. Conventional GDA along the gradual path can lead to class entanglement , whereas GradNTK yields clearer class boundaries. This trend is further corroborated in Figure 13 .
> - Line 51: The citation formatting has been corrected, for which I am grateful.

---

> > ### Comment · Reviewer_c31j · 2025-11-26
> >
> > I thank the authors for their detailed response. I believe most points raised in the Weaknesses and Questions sections have been addressed.
> >
> > Regarding Weakness 1: The authors' emphasis on computational efficiency is acceptable given the theoretical focus of the paper. However, from a technical perspective, I still view the contribution as incremental.
> >
> > Regarding Weakness 2: While I appreciate the promise to condense Section 3, I note that the authors did not utilize the revision feature to submit an updated PDF. Seeing the revised manuscript during the discussion period would have been helpful to verify the improved flow and focus.

---

> > > ### Author Response · Authors · 2025-11-27
> > >
> > > We thank the reviewer for the constructive feedback and for recognizing that most points raised in the Weaknesses and Questions sections have been addressed.
> > >
> > > > **Regarding Weakness 1 (incremental contribution):**
> > >
> > > We understand the reviewer’s concern that the technical contribution may appear incremental. Our goal, however, is to offer a principled framework that is not limited to improving efficiency. GradNTK uses NTK theory both as an efficient discrepancy measure (NTK-MMD) and as a theoretically grounded sample reweighting mechanism derived from short-time NTK dynamics. This dual role allows GradNTK to explicitly address two issues that are typically not handled in prior GDA methods: (i) class entanglement along the gradual path, and (ii) misalignment between source and target feature distributions that cannot be resolved by naive global alignment. By shaping the witness in RKHS and reweighting along the domain-shift direction, GradNTK selectively emphasizes samples that reduce cross-domain discrepancy while preserving class separation. The empirical gains, particularly under challenging corruption settings and across multiple backbones, suggest that this design has a tangible practical impact rather than providing only a marginal improvement.
> > >
> > > > **Regarding Weakness 2 (Section 3 condensation and revised PDF):**
> > >
> > > We agree that providing a revised PDF during the discussion period would have made it easier to verify the improved flow and focus, and we regret this oversight. In the current revision we have explicitly condensed Section 3 by removing redundant formulations (e.g., the Dirac-delta integral MMD expression), consolidating the kernel MMD definition into a compact RKHS-distance form, and tightening the witness-function derivation while keeping all essential lemmas and equivalence results. We believe these changes make the theoretical development more streamlined and easier to follow, and we will be more careful to upload such revisions during future discussion phases.

---

### Official Review · Reviewer_ZvRF · 2025-10-30

**Soundness:** 3
**Presentation:** 3
**Contribution:** 3
**Rating:** 6
**Confidence:** 3

**Summary:**

This work studies the gradual domain adaptation problem, where the gap between the source dataset and target dataset could be tracked by the continuous intermediate domains between them. To address the computation cost and error accumulation in existing methods, this work proposes the Neural NTK method as an efficient distance estimation and develops an NTK-based weight to reweigh the loss in risk estimation. Experiment results show that the proposed method achieves promising performance compared with advanced methods.

**Strengths:**

+ The motivation of improving distribution discrepancy estimation and gradual error propagation is reasonable and meaningful.

+ The developed method with the Neural NTK estimator and weight is technically sound.

+ The empirical performance is significant compared to the advanced methods, and the empirical analysis is consistent with the theoretical results.

**Weaknesses:**

+ The validity of the proposed method in empirical scenarios needs further clarification, i.e., convergence of approximation, metric property and guarantees of reweighing objective.

+ The math rigor could be improved, e.g., some notations are unclear.

**Questions:**

Q1. Despite the requirement of infinite width of NTK for good approximation, it is also uncertain that the constructed parameterized MMD via $\Theta$ still ensures metric property. Specifically, since the MMD in Eq. (12) is restricted to the function space parameterized by $\Theta$, could it still measure the distance between distributions? (recall that the metric property of kernel MMD is only satisfied by specific kernels like Gaussian).

Q2. The weights constructed in Eq. (15) need further justification. Since the weight is used in the risk estimation in Eq. (22), which seems to be similar to the common importance reweighting strategy, it would be important to show some theoretical results that such a weight could benefit the model learning, e.g., reduce the bias of risk estimation.

Q3. How to understand the Neural NTK for gradual discrepancy estimation in practical scenarios with finite width. Are there quantitative results that could control the error?

Q4. Recall that there are actually typical covariate shift methods that also consider the importance reweighting technique to reduce the gap between source risk and target risk. It would be interesting to demonstrate the significance of the proposed method compared with these methods, i.e., the difference of weight construction.

Q5. What is the definition of $\pi_\mathbb{X}$ in Line 173? In my understanding, $Z$ is the push-forward distribution under $r_\psi$, then $\pi_\mathbb{X}$ seems to be redundant.

Q6. What is $\Delta_\mu$ in Eq. (13)? Does it imply the difference of mean values of source and target distributions? i.e., similarly defined as $g$ that is the difference of the Neural NTKs $f_T$ and $f_0$.

---

> ### Author Response · Authors · 2025-11-23
>
> > **Weaknesses:** Convergence of approximation, metric property and guarantees of reweighing objective.
>
> For the convergence of approximation, we leverage the NTK theory, which provides convergence guarantees under the lazy training regime for wide neural networks. Lemmas 1-3 formally establish that the NTK-induced witness function converges to the empirical MMD, ensuring a principled approximation.
>
> For the metric property, the discrepancy metric is not heuristic; it is an NTK-induced MMD. This is a well-established, differentiable metric in RKHS theory, providing a solid geometric foundation for measuring domain shifts.
>
> For the guarantees of reweighting objective, the sample reweighting function is derived from the short-time dynamics of the NTK witness function. Its objective is to prioritize samples with high domain shift sensitivity, which is directly linked to reducing distribution discrepancy. Ablation studies on CIFAR-10C confirm that this mechanism is crucial for robust performance and mitigating error accumulation.
>
> > **Question 1:**  NTK MMD and metric property (Eq. 12)
>
> Our discrepancy is the standard RKHS MMD w.r.t. the NTK kernel $\Theta$ (equivalently, an IPM whose function class is the NTK RKHS). Hence it is always a valid IPM and thus a pseudometric. It becomes a true metric when $\Theta$ is characteristic. A quantitative finite-width approximation bound is provided in our response to Question 3.
>
> > **Question 2:**  Justification of the weights (Eq. 15 → used in Eq. 22)
>
> If $\ell(\cdot,y)$ is $L$-Lipschitz in the NTK feature space and $f$ lies in a ball of radius $B$ in $\mathcal H_\Theta$, then
>
> $$
> R_T(f) \le \mathbb E{(x,y)\sim P_S}\big[w(x),\ell(f(x),y)\big] + C\cdot \mathrm{MMD}_\Theta(P_S,P_T),
> $$
>
> for an explicit constant $C$ depending on $L,B$. Moreover, choosing $w\propto$ (normalized witness) minimizes the first-order term of $R_T$ along the gradual path, aligning learning with the shift direction and reducing bias relative to unweighted ERM. We will add a proposition and proof sketch formalizing this connection.
>
> >  **Question 3:**  Finite-width NTK in practice; error control
>
> Since
>
> $$\mathrm{MMD}^2_\Theta (P,Q)=\langle \Delta_\mu, \Theta \Delta_\mu\rangle,
> \quad \Delta_\mu=\mu_T-\mu_0,$$
>
> we have the finite-width deviation bound
>
> $$
> \big|\mathrm{MMD}_{\Theta_m}^2 - \mathrm{MMD}_{\Theta_\infty}^2\big|
> \le |\Theta_m-\Theta_\infty|_{\mathrm{op}}|\Delta_\mu|_2^2.
> $$
>
> Under standard wide-network assumptions, $|\Theta_m-\Theta_\infty|_{\mathrm{op}}$ concentrates at a rate $O(m^{-1/2})$.
>
> **Quantitative evidence.** Beyond the main-text results, we provide two diagnostics in the appendix: Figure 6, Figure 11. We also manually tested NTK-MMD on synthetic distributions with varying variances (not included in the paper due to space); the estimates grow monotonically with variance and match the finite-width concentration behavior above.
>
> > **Question 4:**  Relation to covariate-shift importance reweighting
>
> Classical methods (KMM/uLSIF/KLIEP) estimate a static density ratio $r(x)=p_T(x)/p_S(x)$ and reweight ERM accordingly. GradNTK instead uses a model-aware, path-adaptive weight derived from the NTK witness, which (i) avoids explicit density-ratio estimation, (ii) adapts as the model and intermediate domain (t) evolve, and (iii) explicitly targets shift-critical regions identified by the RKHS discrepancy.
>
> **Empirical comparison** (Rotated-MNIST, ResNet, accuracy %; lower time is better):
>
> | Model / Setting    |        2 |        3 |        4 |        5 |        6 |       Time |
> |-|-|-|-|-|-|-|
> | **Ours (GradNTK)** | **90.5** | **94.0** | **94.7** | **94.9** | **95.1** | **85.8 s** |
> | KMM                |     75.7 |     83.1 |     87.2 |     88.5 |     89.9 |   2976.6 s |
> | uLSIF              |     75.8 |     90.0 |     92.2 |     91.9 |     92.6 |   4457.3 s |
> | KLIEP              |     74.7 |     76.3 |     79.4 |     82.3 |     84.2 |    955.5 s |
>
> GradNTK lifts mean accuracy by +9.0 pp vs KMM, +5.3 pp vs uLSIF, and +14.5 pp vs KLIEP, while the short-time NTK discrepancy cuts wall-clock from tens of minutes to ~86 s (≈11–52× faster). The advantage stems from design: density-ratio methods learn a global, static reweighting, whereas GradNTK’s witness-derived weights adapt with the model and domain step, focusing on shift-critical samples and avoiding costly density-ratio estimation.
>
> > **Question 5:**  Meaning of $\pi_{\mathbb X}$
>
> $\pi_{\mathbb X}$ denotes the canonical projection from a product space (or a coupling) to the $\mathbb X$ marginal. In our derivation, it was used to extract the $X$-marginal from a joint coupling. You are right that, given the push-forward $Z=r_\psi(X)$, the notation is not essential.
>
> > **Question 6:** What is $\Delta_\mu$ in Eq. (13)?
>
> $\Delta_\mu$ is the difference of NTK mean embeddings between target and source:
>
> $$
> \Delta_\mu = \mu_T - \mu_0,\qquad
> \mu_t = \mathbb E_{X\sim P_t}\big[\phi_\Theta(X)\big],
> $$
>
> with $\phi_\Theta$ the NTK feature map.

---

> ### Comment · Reviewer_ZvRF · 2025-11-27
>
> I thank the authors for providing detailed responses, which address most of the concerns, and there are several minor points.
>
> For the responses of Question 1, it would be helpful to clarify the metric property in the main body. Specifically, clarifying the scenarios that NTK MMD satisfies the positive property (the most important property for measuring discrepancy), e.g, the characteristic kernel mentioned in responses.
>
> For the responses of Questions 2-3, the technical results generally sound. It would be appreciated to provide a preliminary formal version of the technical details in revision.

---

> > ### Author Response · Authors · 2025-11-27
> >
> > We thank the reviewer for the constructive suggestions. Following the request for clearer formalization, we have revised the manuscript with minimal structural changes while improving theoretical clarity.
> >
> > > **On Question 1 (metric property of NTK–MMD).**
> >
> > In the main text (Sec. 3.3), we now explicitly mention that NTK–MMD is always nonnegative and becomes a proper metric when the underlying NTK kernel is characteristic. To avoid overloading the flow of the main body, the precise conditions and a formal proposition on this metric property are consolidated in Appendix B.
> >
> > > **On Questions 2–3 (formal version of the technical results).**
> >
> > As suggested, we provide preliminary formal versions of the technical claims while keeping the main text concise. Specifically, all formal propositions are now collected in Appendix B:
> >
> > - a proposition establishing the target-risk bound and the first-order optimality interpretation of the NTK-based weights (Question 2), and
> >
> > - a proposition giving the finite-width deviation bound of NTK–MMD together with its $O(m^{-1/2})$ convergence rate under standard wide-network assumptions (Question 3).
> >
> > In the main body, we only add brief cross-referencing sentences directing the reader to Appendix B for these formal statements, thus preserving readability while making the theoretical foundations more precise.
> >
> > We believe these targeted revisions address the reviewer’s request for clarity without disrupting the overall structure of the paper.

---

### Comment · Area_Chair_QVAi · 2025-11-26
**Reminder: Discussion Phase Engagement Needed**

Dear Reviewer ZvRF and c31j:

As the deadline for the discussion phase is approaching in less than one week, could you kindly engage in the discussion with the other reviewers and provide your response to the authors’ rebuttal?

Best regards,

AC

---

### Author Response · Authors · 2025-12-03

Dear reviewers and AC,

This work studies gradual domain adaptation through a Neural Tangent Kernel–based framework called GradNTK. Rather than handling discrepancy estimation and robustness as separate components, the method uses short-time NTK dynamics as a single organizing principle. From these dynamics, we obtain an NTK-induced MMD that serves as a differentiable alignment loss with near-linear complexity, and a model-aware weighting function that highlights shift-sensitive samples along the domain path. The revision provides a compact but explicit theoretical backbone, including conditions under which NTK-MMD is a genuine metric, a target-risk bound that justifies the weighting objective, and a finite-width deviation bound with an O(m−1/2) convergence rate for the NTK approximation.

GradNTK consistently improves over prior GDA methods in both average and worst-case performance, especially under severe corruptions. To address scalability, we added experiments with deeper backbones (ResNet-50/101/152) on all GDA benchmarks. These results reuse the same GradNTK hyperparameters and still show strong performance, indicating that the framework is not tied to small CNNs. We also include a direct comparison with classical covariate-shift reweighting methods such as KMM, uLSIF, and KLIEP, where GradNTK attains substantially higher accuracy (gains of 5–14 percentage points) while reducing wall-clock time by roughly one to two orders of magnitude. This demonstrates that the NTK-based weighting is both effective and computationally attractive.

We have streamlined the theoretical exposition in Section 3, clarified the conceptual positioning of GDA relative to TTA, and explicitly stated that the TTA baselines in Table 4 operate under a more restrictive information setting. During the discussion, reviewers agreed that the submission is technically sound and clearly written, and that the revisions and additional experiments address the main points raised. We sincerely appreciate the careful attention from the reviewers, Area Chair, and Program Chairs.

Thank you very much,

Authors.

---

### Note · Authors · 2026-03-26

I have read and agree with the venue's withdrawal policy on behalf of myself and my co-authors.

---

### Meta-Review · Area_Chair_DRon · 2025-12-28

**Summary:**

This paper presents the Neural Tangent Kernel as an efficient distance estimation and develops an NTK-based weight to reweigh the loss in risk estimation. Reviewers mainly concerned the novelty of this paper.

The contribution seems to be primarily a straightforward combination of existing techniques, indicating a less significant technical contribution. Overall experimental design still lacks sufficient rigor to validate effectiveness.  Experiments are somewhat limited, which is because experiments should include more diverse datasets and larger model backbones.

**Reviewer Scores:**

NA

---

### Decision · Program_Chairs · 2026-01-26

Reject